# A system-wide snapshot: A multi-campus survey of open source contributors at the University of California

Virginia T. Scarlett [1,2*], Renata Gonçalves Curty [1], Juanita Gomez [3], Laura Langdon [2], Greg Janée[1], Amber E. Budden [1*]

1 Library, University of California, Santa Barbara, California, United States of America, 2 University of California OSPO Network, California, United States of America, 3 Department of Computer Science and Engineering, University of California, Santa Cruz, California, United States of America

* virginiascarlett@ucsb.edu (VS); aebudden@ucsb.edu (AB)

## Abstract

Academic open source contributors face a wide array of challenges, making it difficult for universities, support staff, and funders to determine which needs they should prioritize. To help address this problem, the University of California (UC) Open Source Program Office (OSPO) Network conducted a multi-campus survey of open source contributors and aspiring contributors. The goals of this survey were two-fold. First, we aimed to understand the needs of university open source contributors, so that we might design programs to address those needs. Second, we sought to characterize open source activity on campus, in order to assess the value of an OSPO. We received 294 valid responses from students, faculty, researchers, and staff. 93% of students and 92% of researchers report that open source software is important for their work. 58% of experienced open source contributors have served as project maintainers, indicating that a large number of university affiliates not only use open source software, they also build and maintain it. The most common challenge is lack of time, particularly time for writing documentation. Regarding opportunities for support, respondents strongly prioritized access to robust computing environments and dedicated grants for sustainability. Finally, comments revealed that institutional norms and priorities can sometimes impede contribution. The survey findings reveal diverse needs across contributor groups, with resources, infrastructure, and culture all playing a role. At the same time, the abundance of maintainers, the prevalence of time and funding-related challenges, and the comments regarding maintenance all underscore a critical need for support for open source sustainability. We conclude by recommending actionable strategies universities can adopt to incorporate sustainability into their open source initiatives. We expect the findings to extend beyond OSPOs to benefit scholars of open source and research software more broadly, providing empirical insights into open source participation and sustainability in academic contexts.

**Data availability statement:** Anonymized and minimally processed data are available on Dryad: https://doi.org/10.5061/dryad.2280gb662. A snapshot of the code at time of publication is archived in Zenodo: https://doi.org/10.5281/zenodo.17783102. The GitHub repository, which may be modified in the future, is available at https://github.com/UC-OSPO-Network/ospo-survey-analysis.

**Funding:** This work was supported by the Alfred P. Sloan Foundation under the grant "Building an OSPO Network at the University of California" (Grant No. G-2024-22424), awarded to A.B. The funders had no role in study design, data collection and analysis, decision to publish, or preparation of the manuscript.

**Competing interests:** The authors have declared that no competing interests exist.

## Introduction

Open-source technology development brings together communities of innovators across the globe while serving a broad and diverse user population. Surveys indicate that 94% of organizations use open source code [1] and 68% of companies increased their use of open source software in 2023 [2]. Broadly speaking, "open source" refers to products for which the code and/or designs are publicly accessible, allowing for modification and redistribution [3]. Many prominent open-source projects—such as Python, R, MySQL, Linux, Jupyter, Firefox, Zotero, WordPress, and Arduino—are widely used in research, business, and education.

Recognizing the strategic importance of open source, many organizations have established Open Source Program Offices (OSPOs) to coordinate their open source policies, strategies, and compliance efforts. The first OSPO to call itself an "OSPO" was formed at Google in 2004, concurrent with, or soon followed by, similar initiatives at Intel and other technology companies [4]. Today, OSPOs are most commonly found at large organizations, but they are rapidly growing in small to medium organizations (fewer than 1,000 employees) [1]. Core responsibilities of OSPOs typically include improving the organization's open source policies, guiding its open source strategy, overseeing license compliance, and advising on open source best practices [1].

Universities have also acknowledged the growing need for open source policies, strategies, and educational programs. The first academic OSPO in the United States was established at Johns Hopkins University in 2019 supported by the Alfred P. Sloan Foundation, and to date the SustainOSS group reports that approximately 25 academic OSPOs exist worldwide [5]. These offices' activities may include advising researchers on funding opportunities, helping the university leverage open source for education and research, and educating the campus community about open source best practices [6].

While academic OSPOs share some similarities with their industry counterparts, universities operate under different incentives and goals. Compared to a company, a university may be more concerned with training students, securing grants, and properly preserving research products in the scholarly record. Additionally, there may be cultural differences between industry and academia–for example, academia has traditionally emphasized open dissemination of knowledge, with free sharing of software and data being a long-standing norm [7–9]. Given that the academy has distinctive norms and goals, the model of the industry OSPO must be tailored to fit the academic environment.

There is substantial existing literature on open source activity, usage, or developer needs from studies employing a range of quantitative and qualitative methods. Previously studied themes include motivations for contributing to open source [10–15], reasons for disengagement [11,16,17] or failure [18], developer collaboration and social dynamics [13,19–22], and sustainability and success of open source [23–26]. Other more comprehensive works explore the history and current state of open-source software research [27,28], providing a general overview of this topic. Several

studies on academic research software address success factors [29,30] and critical issues like training, funding and sustainability [31–33].

In contrast to studies relying on self-reported data, other research efforts utilize programmatic data from software collaboration platforms like GitHub and Software Heritage to analyze open source activity. Examples of these include GitHub's mining studies that discuss usage patterns [34] or systematically map GitHub projects by topic [35]. More recently, other works have focused on systematically identifying universities' open source contributions [36] and research software [37]. Finally, some studies from non-profit foundations have examined university OSPOs [38], while others have focused on industry OSPOs [39], including their organizational and business value [40] and their state and adoption [1].

Two studies are particularly relevant to the present work. One is a survey conducted by the OSPO at the University of Wisconsin, Madison, in 2024 [41]. That survey focused on open source usage, perceptions, and campus culture. To our knowledge, this was the first survey of campus needs from a university OSPO whose results were shared with the public. However, the UW-Madison survey mainly asked about user experiences and did not investigate the specific challenges or gaps that would inform how the campus could provide better tools and targeted support for contributors to open source within the university. Another survey relevant to this project [32] was conducted as part of the conceptualization of the US Research Software Sustainability Institute (URSSI). This survey was broader and more comprehensive, investigating a range of challenges faced by research software users and developers, including practices, tool support, training, funding, career paths, credit, and diversity. However, the URSSI survey was deployed at a national scale, with many questions about the kinds of resources available at the respondents' institutions. Our interest was campus-level in scope, being focused on the kinds of programs a burgeoning university OSPO might offer. Additionally, the URSSI's survey focused on research software development, while we were interested in open source contribution patterns. While much research software is open source, open source is a broader category that may include hobby projects, operational infrastructure projects, and more. Because OSPOs are concerned with all things open source, including its culture and perception among researchers as well as non-researchers on campus, this study focused on the broader category of open source projects, which includes but is not limited to research software.

The University of California (UC) OSPO Network, funded by the Alfred P. Sloan Foundation, is a collaborative initiative comprising six UC campuses (UC Berkeley, UC Davis, UC Los Angeles, UC San Diego, UC Santa Cruz, and UC Santa Barbara) to accelerate knowledge and resource sharing through a systemwide network of OSPOs. The Network's strategy is built on three pillars: Discovery, Education, and Sustainability. In its first year, the UC OSPO Network conducted the survey presented here in order to advance these goals.

The present survey builds on prior studies by providing a comprehensive, practical investigation of open source contributor needs from an academic OSPO perspective. Its primary objectives were to characterize the open source landscape at UC, identify barriers to sustained engagement, gather input on prospective support initiatives, and evaluate the perceived value of an OSPO. The instrument was designed chiefly for experienced contributors, who are well-positioned to offer informed feedback, though it also included a question for aspiring contributors.

Academic OSPOs are a relatively new phenomenon, and investigation and experimentation will be required to discover how these offices can best serve their campus communities. This survey can serve a starting point for other university OSPOs and highlight potential pitfalls, especially around distribution and survey design (see Limitations). The survey and anonymized data are made available to further reuse (see Data Availability). We hope this study will help university leaders and support staff make data-driven decisions that will foster greater engagement with open source by a broader segment of the university community.

## Materials and methods

### Survey design

The survey (S1 File) was developed through a multi-step approach including targeted literature review and iterative item generation. Candidate questions and multiple-choice options, either drawn from prior studies and newly developed by the

research team, were compiled and then selected by group consensus for their alignment with the study goals. Our survey prioritized three domains: contributor needs, open source activity, and perceptions of an OSPO's value. It also invited respondents to provide their GitHub (or GitLab, etc.) usernames, if they were comfortable doing so.

To verify the soundness of the survey instrument, we conducted a pre-test. We sent the draft survey to 11 colleagues at non-UC research institutions. We deliberately tried to select pre-testers from a variety of backgrounds, including DevOps/system administration, technical writing, professorship, research software engineering, and research assistantship. Many of the pre-testers, but not all, had experience working in a university setting. All had at least some experience contributing to open source. We ultimately received feedback from eight pre-testers. All pre-testers were satisfied with the flow and length, and where applicable, all were comfortable sharing their GitHub information. Suggested changes included clarifying the language, specifying how to enter a list of items into the free text boxes (e.g., comma separation), adjusting certain questions to be more inclusive of non-maintainers and hardware projects, and restructuring the question about project sizes to be clearer and more inclusive.

The revised survey was deployed in Qualtrics and consisted of six sections. The survey directed participants to appropriate sections based on their responses (Fig 1). Section 1 of the survey was a consent form. Section 2 of the survey asked participants their affiliation to ensure only UC affiliates completed the survey. Participants could select only one affiliation. Section 3 of the survey asked two introductory questions: one about the importance of open source, and the other about the participant's status as an open source contributor. Experienced open source contributors were then taken

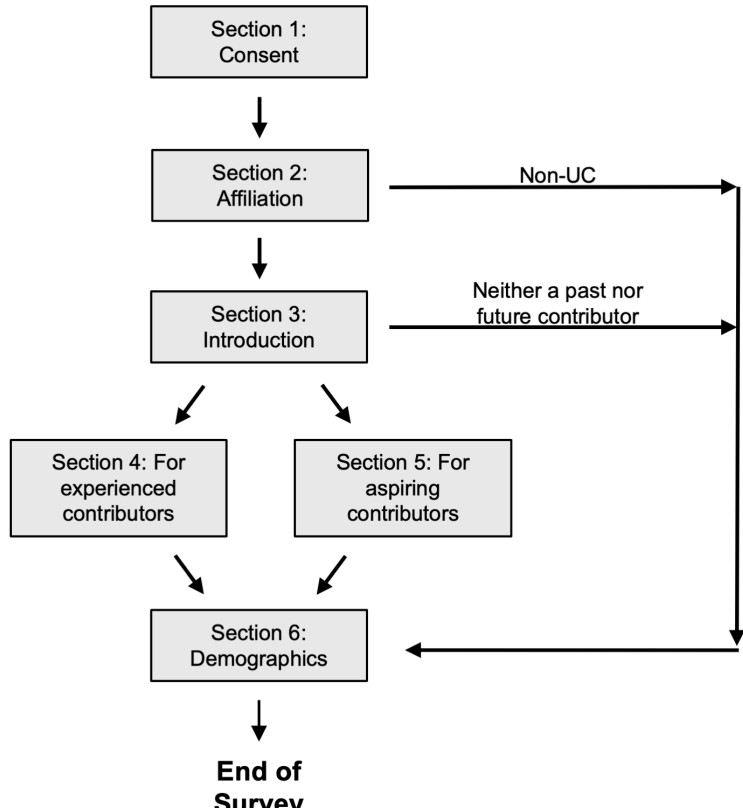

**Fig 1. Survey flow.** Certain participants were immediately directed to the end of the survey: these were participants who were not affiliated with UC, and participants who had neither experience with, nor interest in, open source contribution. Section 5 of the survey consisted of a single question. The bulk of the survey questions were in Section 4 of the survey.

to Section 4 of the survey, a large block of questions that constituted the bulk of the survey. Aspiring open source contributors were directed to Section 5 of the survey, which consisted of a single question about getting started in open source. People who identified as neither an experienced nor aspiring contributor were taken to the end of the survey. Experienced and aspiring contributors were next directed to Section 6 of the survey, which consisted of demographic questions. Section 6 of the survey also asked whether they'd like to join our mailing list. The survey took experienced contributors (the group with the most questions) about 10–15 minutes to complete.

## Data collection

We employed a non-probability snowball sampling approach and a combination of outreach methods. This approach was particularly suitable given the study's focus on accessing contributors whose identities and availabilities were largely unknown and who possessed specific knowledge or experiences not easily identifiable through conventional sampling techniques. We disseminated recruitment materials via multiple channels, including campus-wide and departmental email lists, relevant student group lists, physical flyers on campuses and at events, and professional Slack and Discord channels. While we made some extra efforts to reach Computer Science departments and Information Technology (IT) departments, we by no means limited our distribution efforts to these groups. Due to our limited access to communication channels on other campuses, distribution was largely limited to the six UC OSPO Network member campuses, although a handful of announcements reached the other campuses, through either shared Slack channels or favors from colleagues and friends.

The survey was launched on March 31, 2025, and remained open until May 9, 2025, with up to two reminders, depending on the channel, to increase participation. One week into active data collection, the team amended the question about institutional affiliation to include other UCs not represented in the survey (i.e., Lawrence Berkeley National Laboratory, UC Office of the President, UC Agriculture and Natural Resources, and/or UC College of the Law, San Francisco).

Survey responses were collected anonymously in Qualtrics. Participants had the option to provide their email addresses and/or GitHub usernames for follow-up communications with the OSPO Network. For those who chose to share this information, survey responses were separated from any personal identifiers prior to the analysis process.

## Ethics statement

The survey instrument and protocol were approved by the UC Santa Barbara Office of Research Institutional Review Board Human Subjects Committee (HSC) (protocols 1-25-0182, 1-25-0237, and 1-25-0264), and were deemed exempt from human subjects review, i.e., Federal Regulations 45 CFR 46.104(d), under category 2. The HSC also determined that letters of permission from other campuses were not necessary in this case. Written informed consent to publish survey results was obtained from all participants at the start of the survey. All personally identifiable information, including indirect identifiers, has been removed from the dataset in Dryad. Details on this procedure can be found in the dataset README. The authors declare that there is no conflict of interest.

## Data analysis

**Software used.** Analyses were conducted using R Statistical Software [42]. The R project environment was managed using the renv package manager [43], and we used the tidyverse ecosystem for data management and visualization [44]. Analyses were performed in Quarto notebooks [45]. Most data visualizations were created with ggplot2 [46,47], while specialized visualizations were performed with the treemapify [48] and ComplexHeatmap [49–51] packages. Figure panels were assembled into figures using the patchwork [52] package. Statistical packages used are cited inline below. A complete list of R packages used in this study may be found in the GitHub repository associated with this study as well as the Zenodo code deposit (see Data Availability), in the packages.R file.

**Regression analysis.** Several regression models were constructed using the quantitative dataset. For most regression analyses, 'job category' (i.e., faculty, postdoc, etc.) was included as a fixed effect. In some analyses the original job categories from the survey were grouped and relabeled so that postdoctoral researchers and staff researchers were combined into one group, and undergraduates and graduate students were also combined into another group. Additionally, when appropriate, we combined these groups even further into academics and non-research staff. These groups are somewhat arbitrary, as evidenced by these alternative labeling approaches. Nevertheless, we focused on job category for three reasons. First, we hypothesized that there would be significant differences between the groups. Second, the boundaries between these groups are more clear and objective than many other open source roles, e.g., maintainer/contributor/community manager. Third, conclusions about these groups lead to readily actionable advice for OSPO staff. For example, mailing lists corresponding to these groups already exist.

To confirm that the chosen explanatory variables were appropriate, parameter-rich models were compared to minimal models using likelihood ratio tests, as implemented in the anova() function in the stats package [42]. Only if the likelihood ratio test yielded a p-value below 0.05 was the more complex model used. In cases where p-values were obtained from multiple sets of nested models, False Discovery Rate (FDR) correction was applied using the Benjamini-Hochberg procedure [53]. Non-nested model fits were compared with the Akaike Information Criterion (AIC) [54], and in some cases standard errors of the coefficients were also considered as goodness-of-fit diagnostics.

"Select all that apply" questions (i.e., S1 File Q15, Q6) were analyzed as a set of logistic regression models where each binary outcome, yes/no, indicated whether a participant had selected that response. In other words, each option was modeled and examined independently. These models were constructed with the glm() function from the stats package [42]. Post-hoc multiple hypothesis correction was performed using the Benjamini-Hochberg procedure [53].

For rating scale questions (i.e., S1 File Q5, Q9, and Q10), in each case we constructed an ordinal regression model with a logit link, as implemented by the clmm() function in the ordinal package [55]. Instead of constructing a model for each option within a given survey question, we instead combined all the options into one model, with the option being ranked as a fixed effect, and the individual respondent as a random effect. After choosing a model as described above, the emmeans package [56] was used to extract estimated marginal mean (least squares mean) ratings and pairwise contrasts, which allowed us to compare ratings within groups, between groups, and globally by holding groups constant. Emmeans was run with mode = "mean.class", meaning the package estimated a probability distribution for each outcome level, and then reported the average of these probability distributions on a scale of 1 to $N_{outcome categories}$. Hence if the survey options were "Not very useful" (1) / "Useful" (2) / "Very Useful" (3), the package estimated the probability distribution for each selection (1, 2, and 3,) and then reported the means of the distributions on a scale of 1–3.

When averaging across job categories (faculty, postdocs and staff researchers, students, and non-research staff) to obtain marginal means, we weighted the job categories equally, effectively correcting for sample sizes. This approach diminishes the influence of well-represented groups (e.g., non-research staff) on the marginal means, while amplifying the influence of less-represented groups (e.g., students). Since the true representation of open source contributors among campus groups is unknown, any attempt to create post-hoc weights based on known demographic data was deemed speculative and potentially unreliable, hence equal weighting was preferred.

Goodness-of-fit testing with ordinal's nominal_test() and scale_test() on the corresponding non-mixed models suggested deviations from the proportional-odds and/or constant-scale assumptions in several cases. We proceeded with caution, restricting our analysis to means rather than slopes, and largely using the regression models for exploratory analysis and hypothesis generation. Notable claims about significant differences between means were confirmed with non-parametric tests. In most cases, a Kruskal-Wallis rank sum test [57], as implemented by the stats package [42], was used with p-value adjustment using the Benjamini-Hochberg procedure [53] as appropriate. For Q9, the Kruskal-Wallis test was followed by a post-hoc analysis of which groups were different with a Dunn's test [58,59]. For Q5, a Wilcoxon rank sum test [60] was used, since only two groups were being compared (small projects vs. large projects).

                                                                    

## Other statistical analysis

To test for differences between proportions (e.g., proportion of hardware contributors among faculty vs. students), a 2-sample z-test for equality of proportions was applied. Prior to each z-test, a post-hoc power analysis was performed using the pwr.2p2n.test() function from the pwr package [61] to confirm the sample size was adequate to achieve 80% power at a significance level of $\alpha = 0.05$.

The Cochran-Armitage test for trend, which was applied to Q6, was implemented with the prop.trend.test() function from the stats package [42] using default parameters.

For Q9, we clustered the response distributions for listed "challenges" as follows. First, the number of responses for each frequency level and challenge combination was tallied. Then the fviz_nbclust() function from the factoextra package [62] and the silhouette() function from the cluster package [63] were used to generate elbow and silhouette plots. Plots were manually inspected, and average silhouette widths were compared, for different k (cluster number) values. These diagnostic tests with both the Partitioning around Medoids (PAM) [64] and k-means [65,66] clustering methods clearly favored k = 3 clusters. K-means clustering with the stats package [42] and PAM clustering with the cluster package [63] yielded the same three clusters. Finally, we bootstrapped mean Jaccard coefficients [67] for the clusters using the cluster-boot() function from the fpc package [68].

## Qualitative analysis

Our procedure for qualitative data analysis of questions Q6 and Q12 was inductive and inspired by the QUAGOL framework [69]. We received 174 valid responses to Q18 about academic discipline (n = 188 academics), 41 valid responses to Q12 about additional challenges and solutions (n = 233 experienced contributors), and 28 valid responses to Q6 about motivations for contributing (n = 233 experienced contributors). Themes emerged from comments, and were iteratively refined and then assigned to responses using Taguette [70]. For Q12, themes were organized into three overarching categories: resources, culture, and infrastructure.

Academic participants were asked to write in their academic discipline for Q18. Their free-text responses were classified according to the Digital Commons taxonomy of academic disciplines [71]. The taxonomy is three-tiered: e.g., Level 1: Medicine and Health Sciences, Level 2: Medical Sciences, Level 3: Neurosciences. A simple fuzzy string-matching algorithm (Levenshtein distances) was first applied to compare responses with taxonomy items. Candidate matches were then manually reviewed, and responses were retained, dropped, or reassigned until each response was linked to at least one taxonomy item. Seventeen participants (10%) listed multiple disciplines, all of which were retained in the final list. We modified the taxonomy in only one respect: in the original taxonomy, the only category pertaining to artificial intelligence (AI) or machine learning (ML) was "Artificial Intelligence and Robotics". We did not use this item, and instead created a new item for just AI/ML, while sorting robotics-related responses into the taxonomy's existing separate "Robotics" item.

## Results

### Demographics

Our first step was analyze the respondents' attributes. This was done in order to contextualize the survey results and highlight potential biases, rather than to make generalizable claims. A total of 294 valid responses were received from students, faculty, and staff affiliated with UC. This population comprised 233 self-declared experienced contributors—individuals who have previously contributed to open source—and 61 "aspiring" contributors—individuals who indicated interest in future participation in open source, but who had not yet contributed to an open-source project (Fig 2A). Aspiring contributors were present among all job categories–faculty, postdocs and staff researchers, students, and non-research staff (S1 Fig). However, the proportion of aspiring contributors was significantly higher among students (41.2%) than among other

groups (18.9% for non-research staff, the group with the next-highest proportion) (two-proportion z-test, one-sided 95% CI lower bound = 0.0835, p = 0.00214, Cohen's h = 0.493).

Due to our snowball survey distribution method (see Materials and methods), participation from campuses was highly varied (Fig 2B). Nearly half (47.3%) of the 294 respondents were affiliated with either UC Santa Barbara (n = 76) or UC Los Angeles (n = 63), and two campuses (UC Merced and UC Irvine) had fewer than ten responses (n = 8 and n = 2, respectively). We received a substantial number of responses (n = 19) from "Other UC", a category that may have encompassed participants from Lawrence Berkeley National Laboratory, UC Office of the President, UC Agriculture and Natural Resources, or UC College of the Law, San Francisco. There was no correlation between the number of responses received and the size of the campus population (S2 Fig).

Academics, which included students, faculty, and staff researchers, constituted approximately two-thirds (63.9%) of the total respondents (Fig 2C, Fig 2D). Response rates varied with career stage. Only 7 experienced open source contributors identified as undergraduates, and only 15 as postdoctoral researchers. Academic respondents (n = 188) were predominantly from STEM fields (86.2%), with mathematics and computer science together accounting for just over half (53.1%) of the STEM-affiliated participants (n = 162) (Fig 2C). At the broadest taxonomic level (see Materials and methods), STEM fields dominated, though Social Sciences showed greater representation based on the write-in responses (13%, n = 174) compared to the multiple-choice responses (8%, n = 188) (Table 1 in S1 Data, Fig 2C). These figures are lower than the 18% representation of Social Sciences that would be expected based on UC enrollment data [72], though it cannot be said whether this difference is "significant". At the second level, Computer Sciences (22%) and Medical Sciences (10%) led (Table 2 in S1 Data), while at the third level, Neurosciences (8%) and AI/ML (4%) were most common (Table 3 in

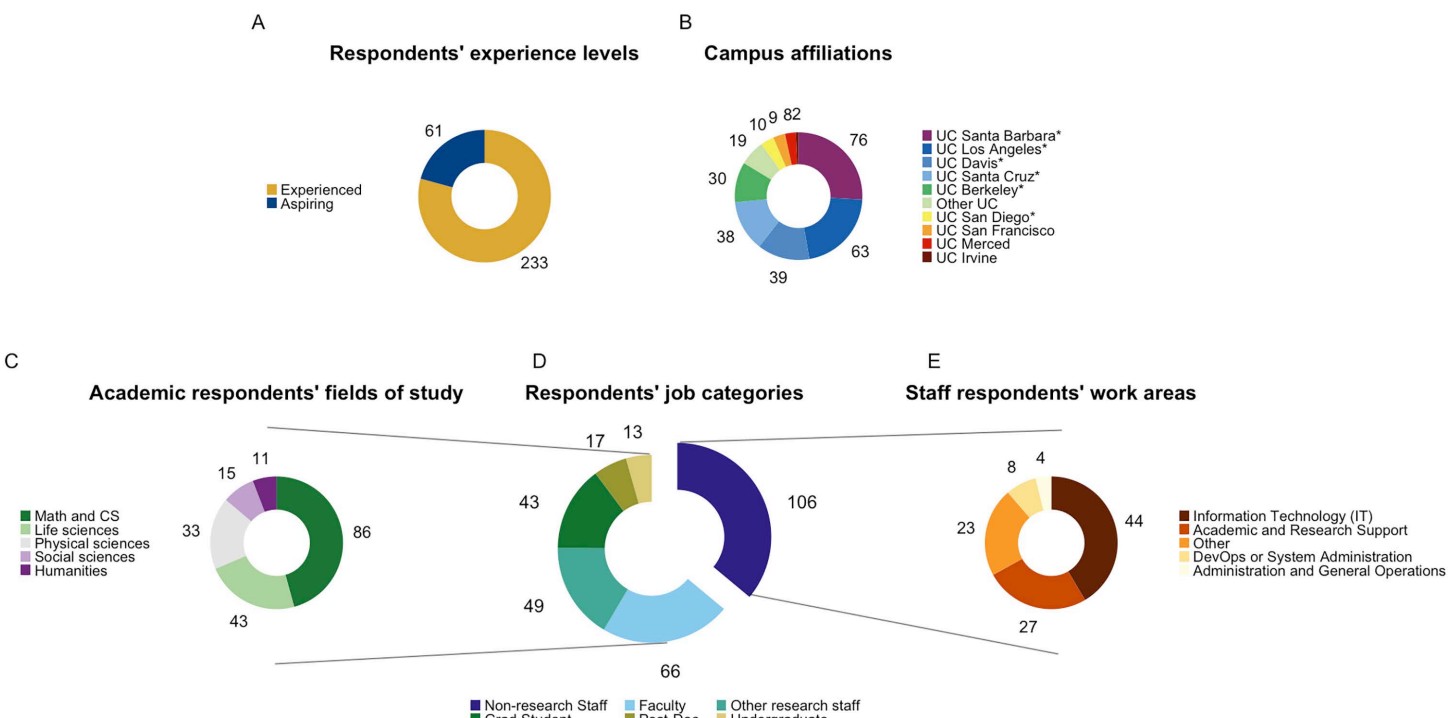

**Fig 2. Demographics. (A)** Proportion of experienced to aspiring contributors. **(B)** Number of responses from each campus. "Other UC" may include the UC Office of the President, Lawrence Berkeley National Lab, UC Agriculture and Natural Resources, and UC College of the Law, San Francisco. Asterisks denote UC OSPO Network member campuses. **(C)** Disciplines of academic respondents (students, teachers, and researchers). **(D)** Job categories of all respondents. **(E)** Work areas of respondents who identified as non-research staff.

S1 Data). A total of 47 level-three disciplines were represented. Classification of disciplines is inherently challenging, given the fluid and interconnected nature of contemporary knowledge [73]. Nevertheless, these results indicate that a wide range of disciplines is represented among open source contributors at UC, though the majority come from STEM disciplines.

About one-third of total respondents (36.1%) were non-research staff, representing the largest of all job categories (n = 106) (Fig 2D, Fig 2E). The majority (67.0%) of this group worked in either Information Technology or Academic and Research Support—a category encompassing research administration, library science, and instructional design (S1 File Q19, Fig 2E). Consistent with high representation of these fields, 6 of the 13 staff respondents who selected "Other" and specified their professional area explicitly mentioned Library or IT roles. Roles with only one or two respondents are grouped into "Other" in Fig 2E, and include fields such as Marketing and Communications, Finance, and Human Resources. It is not known whether the high representation of IT and library staff is due to particularly high interest from those groups or simply because those groups are large, because the UC public data portal does not publish workforce composition data in their entirety [74].

## Importance of open source

Participants were asked about the importance of open-source tools for various types of work they do. This was the only survey question that focused on usage of open source rather than contribution to open source. All of the following percentages exclude "Non-applicable" responses. 79.9% of respondents said open-source projects are above moderate importance ("Important" or "Very Important") for their learning, and 82.3% of respondents said this for their professional development. 79.3% of teachers said that open source is above moderate importance for their teaching, and 92% of researchers said that open source is above moderate importance for their research. (These two groups were inferred based on the participant's election to give a response other than "Non-applicable" for teaching or research, respectively; the teacher group constituted 193/294 or 66% of total respondents, and the researcher group constituted 239/294 or 81%.) 88% of non-research staff (S1 File, Q16) said that open source is above moderate importance for their job (Fig 3, Table 4 in S1 Data), and 93% of students (S1 File, Q16) said that open source is more than moderately important for their learning. These results show a high degree of reliance on open-source tools across the participant population.

**Table 1. Themes from comments on motivations for contributing to open source.**

| Theme | Number of comments |
|---|---|
| Profit-driven corporations | 7 |
| Academic values | 6 |
| It's good for society | 5 |
| It's part of my job* | 5 |
| It's good for my career | 3 |
| It saves money | 2 |
| It improves the tools* | 2 |
| To contribute to research | 2 |

Participants were allowed to select "Other" and write a response to the question, "Why do you contribute to open source?". Themes from coding analysis are shown here. Each comment may reference multiple themes. Themes with an asterisk are very similar to options that were already presented in the multiple-choice portion of the question.

**Table 2. Selected comments from the "culture" category.**

| Comment | Code(s) |
|---|---|
| "If every open source project had a breakdown of what code is where it would make contributing much easier" | Code review/replication/cleaner code |
| "Administrators often fail to understand the value their organization receives and the responsibility to contribute back in kind when their organization uses open source software." | Value of open source tools; University leadership, norms, and priorities |
| "there is a lot of demand on our campus for open source 101 resources from students... how does one get started with the necessary tools, how does one pick a project to contribute to, what are the best practices, etc" | Open source education and careers |
| "I work as a full time software engineer and lecturer in [department]. It is astonishing to me how bad the open-source infrastructure is at my university. We should all be using Github Classroom and have an institutional license to Github. While I am very knowledgeable, it is close to impossible to get help from TAs with cloud computing, class Github administration and creating Docker containers... TAs and students should be learning these technologies and should have a support system to do so…" | Open source education and careers; Computing environments; Skilled personnel; University leadership, norms, and priorities |
| "The University's involvement in open source has been mixed at best, at times harmful. In conversations like this, I see a motivation to provide assistance in areas that we, the open source developers do not need assistance, such as project management, licensing, etc... The university's best support for open source would be to help identify funding sources, provide time to identify and recruit motivated researchers with the technical skill and academic knowledge to build these projects, and provide time and support to continue developing these tools…" | Funding open source projects; Lack of time/"extra" work; Licensing; Maintenance; Skilled Personnel; University leadership, norms, and priorities |

**Table 3. Proposed solutions from comments throughout the survey.**

| Proposed Solution | Number of proposers |
|---|---|
| Internal, possibly centralized personnel for maintenance of open source projects | 4 |
| Code review and replication | 3 |
| Clear, simple guidelines for UC employees who want to contribute to open source | 3 |
| Education about open source careers | 2 |
| A committee to evaluate closed source vs. open source vendors | 1 |
| A catalog of open source software used by UC IT | 1 |

"Number of proposers" indicates the number of people who proposed the idea, in their own words. Proposals that were already mentioned in the survey instrument are not shown here.

## Contribution patterns

93.1% of experienced open source contributors reported that they have held multiple roles. On average, each person held about 4 roles (mean=4.28, median=4, mode=4). 95.7% of the experienced contributors identified as at least one of Maintainer, Contributor, or Bug Reporter, and 50% of those respondents in turn identified as all three (S3 Fig). See S1 File Q4 for the definitions of these terms in the context of this question. 57.5% of experienced contributors identified as a current

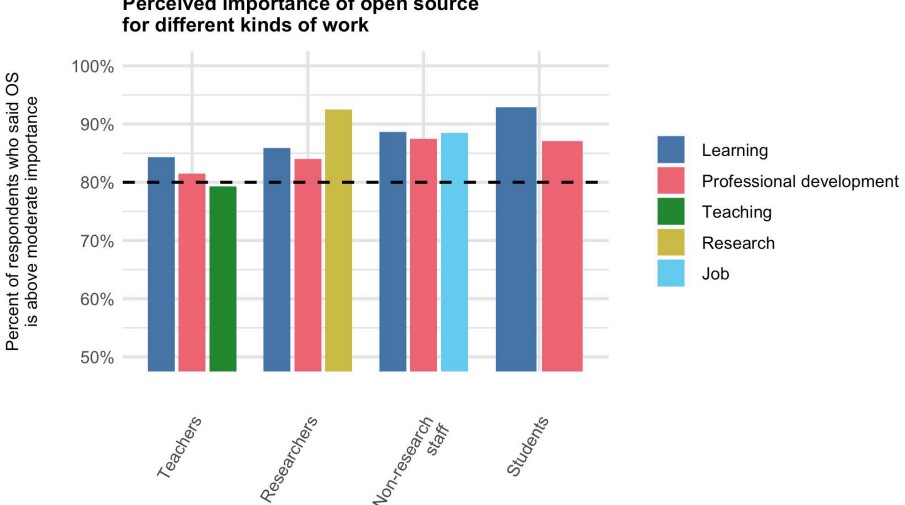

**Fig 3. Perceived importance of open source.** Percent of eligible respondents who reported that open source tools are either "Important" or "Very important" for various tasks. Eligibility was determined by filtering out respondents who selected "Non-applicable"; additionally, for the "Job" task, only non-research staff (S1 File, Q16) were eligible.

or former maintainer. The proportion of respondents who were maintainers varied by campus, from 48.3% to 78.9%, when considering only those campuses that had at least 10 respondents as an adequate sample size. Maintainers outnumbered other contributor roles—community managers, UI/UX designers, and IT/systems administrators—by about 3–1 (134:44, 134:46, and 134:49, respectively). Out of the 9 roles we presented, the mean number of roles held by participants in each role ranged from 4.68 to 6.39. The job category with the lowest proportion of maintainers was undergraduates (42.9%), and the job category with the highest proportion of maintainers was "Other research staff" (Fig 4), a group that included research scientists and software engineers (72.5%) (S1 File Q16). These results indicate that our survey pool contains a relatively high number of maintainers, but also that the lines between roles are not necessarily clear-cut.

We next asked experienced contributors how frequently they contribute to small, medium, or large projects, hypothesizing that small projects might predominate but that there might be differences among job categories. We asked them to answer in terms of relative frequency–that is, which size of project they contribute to most or least often. The data suggested a preference for small projects, with 47% of participants reporting that they contribute to small projects relatively frequently, and that percentage declining to 23% and 16% for medium and large projects, respectively. Ordinal regression without any fixed effects confirmed that for a given participant, the frequency of contribution is significantly lower for medium and large open-source projects compared to small projects. Specifically, the odds of a participant moving into a higher contribution frequency category drop by 69% for a medium project compared to a small project (OR = 0.31, $p < 0.001$), and by 87% for a large project compared to a small project (OR = 0.13, $p < 0.001$) (S4A Fig).

Next, we looked for differences between job categories. Exploratory analysis suggested that all job categories showed similar patterns of contributing to small projects, but some apparent differences emerged for large projects (Fig 5, S5 Fig, Table 6 in S1 Data). To examine these differences more closely, we added job category to our ordinal regression model (S4B Fig). From analysis of the estimated means (Table 7 in S1 Data, Table 8 in S1 Data, Table 9 in S1 Data), we found that all groups had higher odds of contributing to small projects than to large projects (Table 8 in S1 Data). Paired Wilcoxon signed-rank tests within each job category corroborated this finding (Table 10 in S1 Data). We also found that non-research staff report contributing to large projects at a higher frequency than academics (Table 9 in S1 Data), again corroborated by a Wilcoxon test (one-sided, difference = 1, 95% CI lower bound = 0.000025, $p = 0.00003$). In summary, all

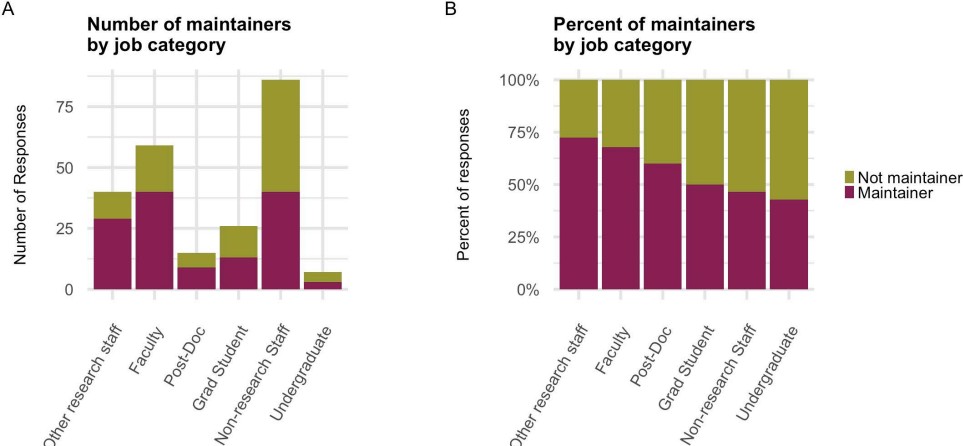

**Fig 4. Proportion of maintainers in each job category. (A)** "Maintainer" indicates the number of experienced contributors in that job category who report having ever been a maintainer of an open source project. **(B)** Same data as in **(A)**, but scaled to 100%. Participants could select multiple roles in addition to maintainer, but they could only select one job category.

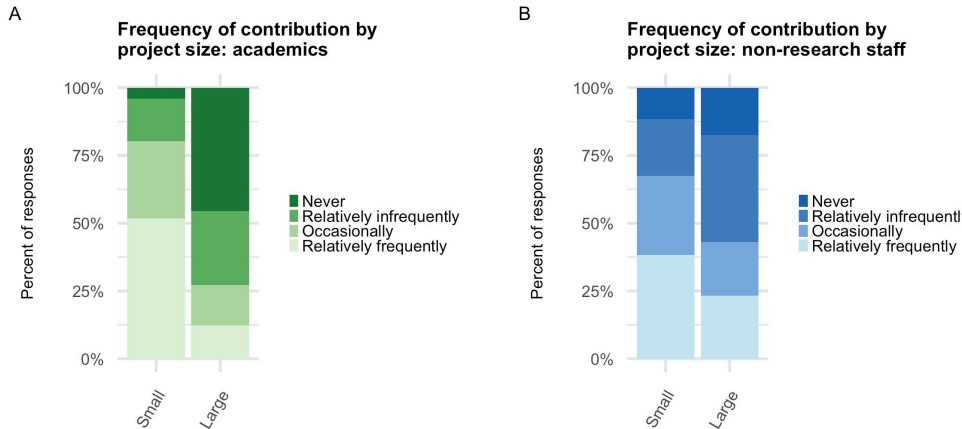

**Fig 5. Frequency of contribution to large vs. small open source projects. (A)** Percent of academics (students, postdocs, research staff, and faculty) who selected each frequency for each size. **(B)** Same as **(A)**, but for non-research staff.

groups tend to contribute to small projects more frequently than large projects, but contributions to large projects are more common among non-research staff.

We also asked respondents about the types of projects they contribute to. Respondents could select all that apply. The aggregate results are in Table 11 in S1 Data. The top two most common responses were "Libraries, packages, or frameworks" and "Applications", with these two receiving almost the same number of responses (157 and 156, respectively, or 67%). On average, each respondent has contributed to about 3 different types of projects (mean number of projects = 2.82, median = 3, mode = 3). We noticed that a larger proportion of students contribute to hardware projects than other groups (S6 Fig). 38.5% of students selected this option, while only 13.6% of faculty–the next-highest rate–selected it, a statistically significant difference (one-sided two-sample z-test for proportions with the next highest group, faculty: $p = 0.0107$, 95% CI (for the difference) = [0.0481, 1], Cohen's $h = 0.584$). This could be due to sampling bias. 6 of the 11 the students who selected "Hardware" came from one campus (UCLA), although 5 of those 6 each identified with a

distinct subfield of study, with three broad domains of study among them (Math and CS, Physical Sciences, and Social Science). This neither proves nor disproves the possibility of sampling bias among student respondents who are hardware contributors; we believe more data are needed to conclude with confidence that students, proportionally, contribute to hardware projects more than other groups.

Finally, we asked experienced open source contributors where they have shared their code and/or hardware designs. 95.3% of respondents (n = 222 out of 233) report having used GitHub to share code and/or hardware designs (Fig 6A). The next most common answer was "A custom website (e.g., a lab website)" (30.5%), which was nearly tied with GitLab (29.6%). When we limited our data to academics (students, teachers, and researchers), the percent of respondents who selected this option remained about the same (32%). The affirmative response rate for "A custom website" varied by academic discipline, with those in the Social Sciences selecting this option with the lowest frequency (20%, n = 2 out of 10), and Math and Computer Science choosing it with the highest frequency (40.3%, n = 29 out of 72) (S7A Fig). Even though GitHub is by far the most popular choice for sharing code, we suspected that many academics also share their code and/ or hardware designs in scholarly data repositories, so we included these as options (S1 File Q8, Fig 6B). Zenodo was the most popular choice among data repositories, with 18.4% of academics (n = 27 out of 147 academics who are also experienced contributors) selecting this option. The affirmative response rates for the repositories varied by UC campus, and these are shown in S7B Fig and S7C Fig.

## Motivations for contributing

Experienced open source contributors (n = 233) were asked why they contribute to open-source projects. Participants were invited to select all that apply from seven motivations. The most popular responses, in order, were: "To improve the tools in my field" (85%), "To customize existing tools to my specific needs" (69.1%), and "To give back to the open source community" (67%) (Fig 7A). These results indicate a focus on improving tools, for a combination of both altruistic and self-interested reasons.

We hypothesized that a participant's motivations for contributing to open source might vary by job category. To test this, we constructed a logistic regression model for each motivation, with job category as a fixed effect, and compared it to the intercept-only model via a likelihood-ratio test (see Materials and methods). Only "To improve my skills" showed a significant job category effect (adjusted p = 2.84 × 10$^{-4}$). Visual inspection of the data suggested that academic participants from

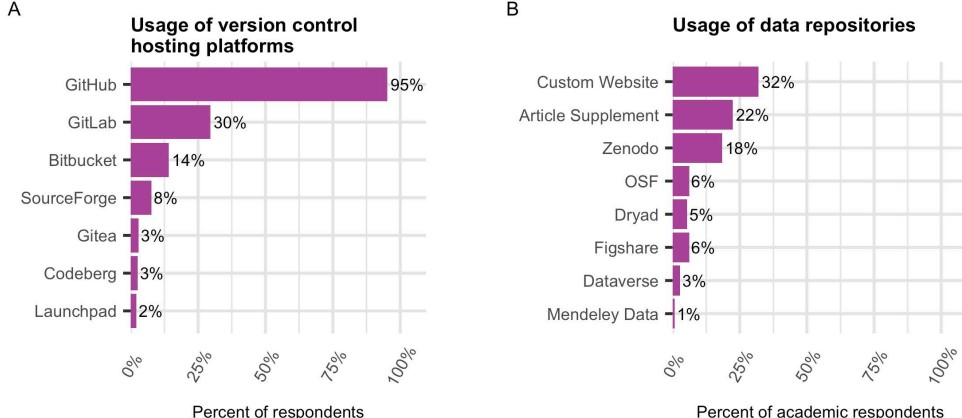

**Fig 6. Code/design hosting platforms. (A)** Percent of total respondents who report ever having used various version controlled platforms to share code and/or hardware designs. **(B)** Percent of academic respondents who report using various data repositories to share code and/or hardware designs.

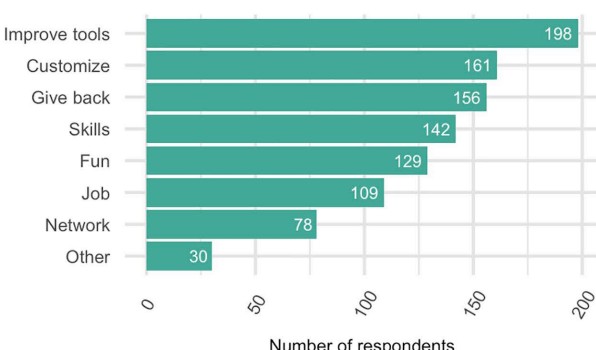

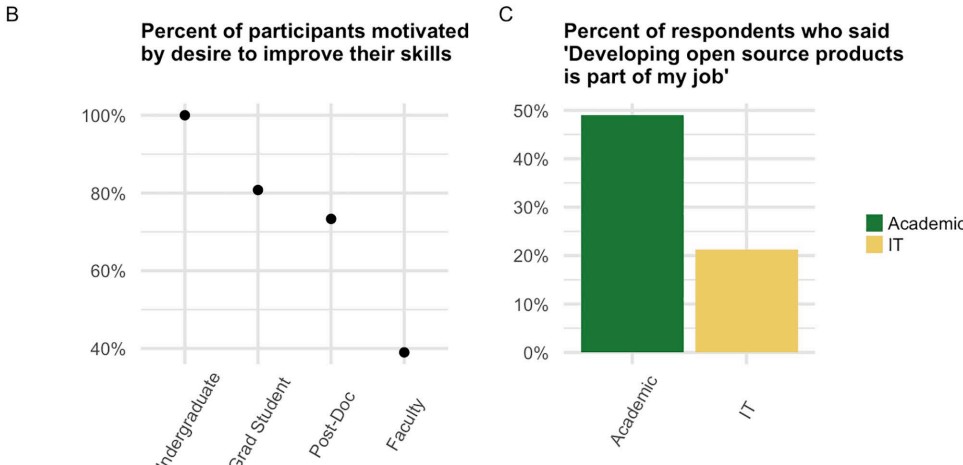

**Fig 7. Motivations for contributing to open source. (A)** Total number of respondents who selected each motivation for contributing to open source. For full descriptions of each motivation, see S1 File Q6. **(B)** Percent of total undergraduates, graduate students, etc. who selected "To improve my skills" as a motivation for contributing to open source. **(C)** Percent of academics (students, teachers, and researchers) and IT staff who reported that they contribute to open source because it's part of their job.

later career stages were less likely to select this motivation, and indeed, a Cochran-Armitage test for trend confirmed this ($p = 8.52 \times 10^{-6}$) (considering the job categories undergraduate students, graduate students, postdocs and staff researchers, and faculty, in that order) (Fig 7B). Taken together, these results indicate that the only motivation that significantly varied by one's job category was the desire to improve one's skills, which was less prominent among academics who are further along in their career.

We hypothesized that IT staff (a subset of non-research staff) and academics (students, teachers, and researchers) might contribute to open source for different reasons, so we applied the same approach as above, but this time the job category variable had only two possible values: IT staff or academic. (Non-IT non-research staff, e.g., librarians, were excluded.) Only for "Developing open-source products is part of my job" did the IT staff vs. academic distinction improve the model (adjusted $p = 0.019$). In our sample population, 49% of academics ($n = 147$ academics who are experienced contributors) vs 21.2% of IT staff ($n = 33$ IT staff who are experienced contributors) selected this motivation (Fig 7C). Consistent with this, a model with just these two job categories found that academics had significantly higher odds of selecting

this motivation than IT staff (OR = 3.57, 95% CI [1.53–9.38], p = 0.00537). A comparable model with non-research staff in place of IT staff was not significantly improved by inclusion of job category (adjusted p = 0.442), indicating that the discrepancy was limited to IT staff.

Finally, we examined the qualitative responses to this question from participants who selected the "Other" option. We coded the responses as described in Materials and methods, and found that eight themes emerged from the data (Table 1, Table 12 in S1 Data). Of these eight themes, two had a relatively high number of comments and were clearly distinct from the options we presented. These were: "Profit-driven corporations" and "Academic values". Comments in the former category referred to "uncaring corporations" and "problematic vendors", and characterized open source as "undermining" or "pushing back" against "capitalist overreach". One participant said they contribute to open source "to stick it to the man." In the "Academic values" category, participants referred to academic ideals or open science in general, and/or referenced the following specific values: autonomy, broad distribution of findings, reproducibility, transparency, and collaboration. The emergence of these two themes suggests that our original list of options did not account for two important motivations–open source contribution as a type of social activism and as an expression of academic ideals.

## Challenges in open source

We presented experienced open source contributors with a list of common challenges in open source, and asked how frequently they encounter each challenge (S1 File Q9). We coded the rating scale as follows: "Never" = 0, "Non-applicable" = 0, "Rarely" = 1, "Occasionally" = 2, "Frequently" = 3, "Always" = 4. The highest-scoring challenges were "Limited time for writing documentation", "Limited time for writing new code", and "Finding time to educate myself" (Table 13 in S1 Data). (Note that for ease of reading, throughout this work, we refer to direct quotations of the survey in double quotes, and shorthand in single quotes.) Thus, the top 3 most frequent challenges all pertain to a lack of time. 'Documentation time' was the most common challenge, with 76% of experienced open source contributors reporting that they encounter this challenge "Always" or "Frequently". The next two most frequent challenges were 'Managing issues' (e.g., GitHub issues) and 'Attracting users'. Focusing on maintainers (n = 134 total maintainers), we found that 49.3% and 47.0% of maintainers said that they encountered these two challenges "Frequently" or "Always". The top five challenges per job category, according to two calculation methods, are shown in S8 Fig.

We observed that the distribution of responses varied, with some challenges having similar distributions. To characterize this more rigorously, we clustered the challenges by their responses (Fig 8) and resolved to three clusters (see Materials and methods). Both k-means and PAM clustering with k = 3 yielded the same clusters, with reasonable quality and stability (average silhouette width = 0.47, clusterwise Jaccard bootstrap means [0.799–0.890]). Challenges in cluster 1 (Fig 8A) received a low number of "Non-applicable" responses and a high number of "Frequently" and "Always" responses; thus, we might characterize these as widespread and frequent challenges. These were the same time-related challenges that received the highest points scores, above, so we call this cluster "Time". Challenges in cluster 2 (Fig 8B) were related to funding and hiring, so we call it "Resourcing". Like the Time cluster, these have a high number of "Frequently" and "Always" responses, but unlike the Time cluster, they have many "Non-applicable" responses. This leads us to interpret the Resourcing cluster as challenges that are not as widespread in terms of the number of people who identify as directly engaging with them, but they are frequent challenges for those who do. All other challenges fell into cluster 3 (Fig 8C), which we call "Other", and which we interpret as moderately widespread and moderately frequent.

We hypothesized that faculty might be particularly likely to select "Frequently" or "Always" for the Resourcing cluster challenges ("Identifying potential funding sources for my open-source projects", "Securing funding for my open-source projects", and "Finding and hiring qualified personnel"). To test this, we constructed an ordinal regression model and estimated marginal means (S9A Fig). We found that for all three of these challenges, faculty gave a higher frequency rating than did students or non-research staff (S9B Fig). All other pairwise comparisons were non-significant. For added confidence, we also conducted a Kruskal-Wallis rank sum test, followed by a Dunn's test, to evaluate whether these groups

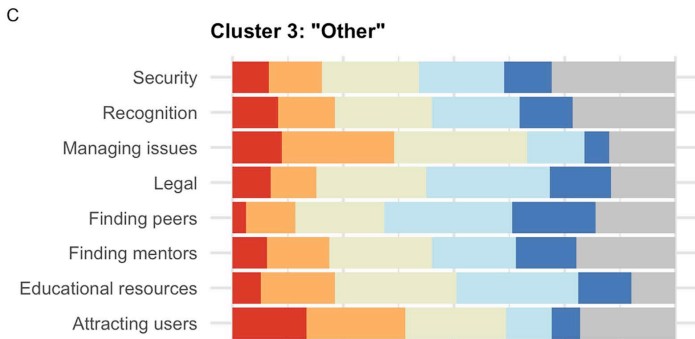

**Fig 8. Perceived challenges in open source.** Reported frequency of various challenges, clustered by the distribution of responses. **(A)**, **(B)**, and (C) respectively show the three clusters identified by both k-means and PAM clustering.

gave different distributions of ratings (see Materials and methods). These tests also indicated that the ratings for these challenges were different between faculty and students/non-research staff (S9C Fig).

## Desirable solutions

To determine what kinds of services academic open source contributors might appreciate and benefit from, we asked respondents to rate the utility of various solutions on a scale of "Not very useful" / "Useful" / "Very useful". We again coded the responses from categorical to numerical outcomes, as described in the previous section. Based on these ratings, the top four most popular solutions, in order, were 'Sustainability grants', 'Help finding funding', 'Computing environments', and 'A learning community' (S10A Fig, Table 14 in S1 Data). A follow-up question asked respondents to select the one solution that would be **most** useful to them. The top four most popular solutions from this follow-up question were the same, but in a different order of importance: 'Sustainability grants' (70 votes, 30% of experienced contributors), 'Computing environments' (39 votes, 17%), 'A learning community' (27 votes, 12%), and 'Help finding funding' (21 votes, 9%) (Fig 9A). By asking the same question in two ways, we gain additional confidence that these four solutions are in particularly high demand.

We also examined which solutions were preferred by each job category. In Q10 (rating scale), 'Sustainability grants' was the highest rated solution for all groups. In Q11 ("choose one"), the same was true for all job categories except

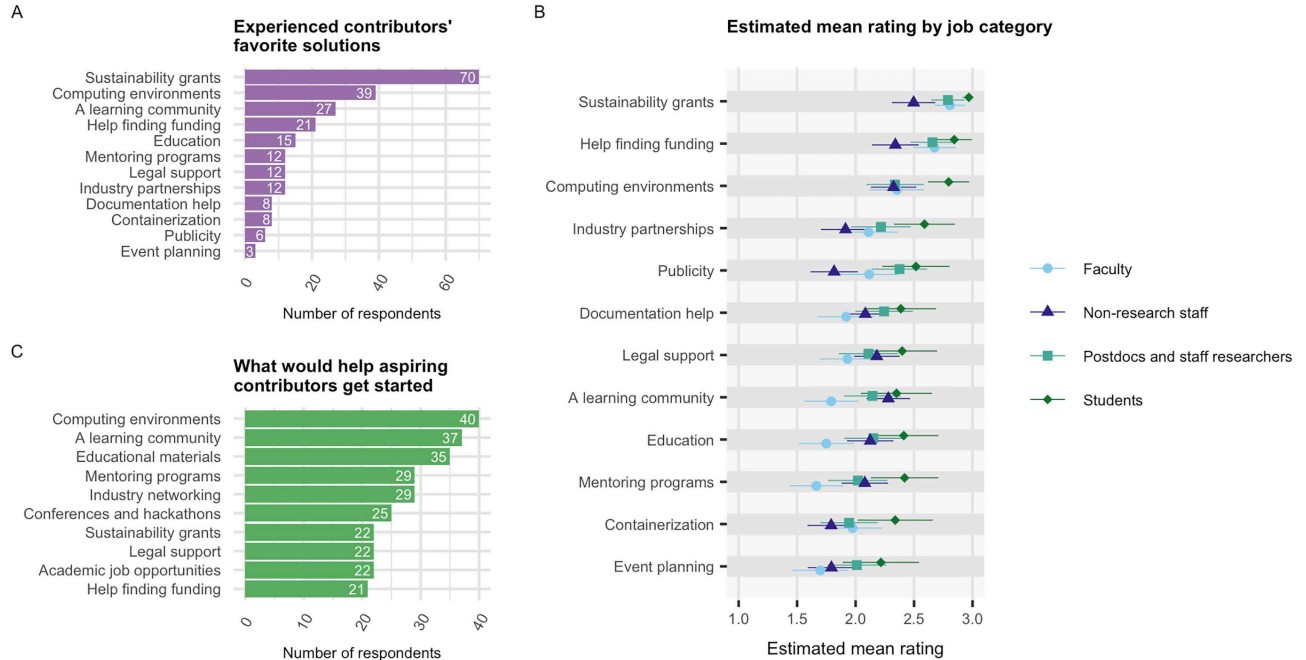

**Fig 9. Desirable solutions in open source. (A)** Number of experienced contributors who voted for various solutions. Respondents were required to choose one. **(B)** Mean perceived utility of various possible solutions, as determined by ordinal regression. 1 = "Not very useful", 2 = "Useful", and 3 = "Very useful". Order of solutions on vertical axis reflects global average ranking when job groups are weighted equally. **(C)** Number of aspiring contributors who voted for various solutions. Respondents were instructed to select all solutions that would be helpful to them.

non-research staff. When non-research staff were forced to choose their most useful solution, 'A learning community' was the most common choice (S10C Fig).

To test the robustness of these findings at a population level, we created an ordinal regression model from the rating scale question (S1 File Q10) (see Materials and methods). The model formula, coefficients, and other model statistics are in S11 Fig. First, we estimated the "global" marginal mean ratings while holding job category constant (Table 15 in S1 Data). We chose to weight the job groups equally. These global ratings are an average across diverse groups. The two funding-related solutions had the highest mean ratings, followed by 'Computing environments'. 'A learning community' was substantially lower on the list than might be expected from the descriptive statistics—ranked 8 out of 12. This is likely due to our weighting scheme. Bolstering this hypothesis, when we calculated the marginal means with weights proportional to sample size, 'A learning community' ranked 5 out 12—three ranks higher (Table 16 in S1 Data). In summary, the model corroborated the descriptive statistics for three of the four top solutions. 'Sustainability grants', 'Help finding funding', and 'Computing environments' were consistently popular, while the popularity of 'A learning community' was tied to the influence of non-research staff.

Pairwise contrasts between the estimated mean ratings (Table 17 in S1 Data) indicated that for nine of the twelve solutions, there was a significant difference in the mean ranking between at least one pair of job categories. In other words, only 25% of the solutions received similar ratings across all groups. To corroborate this finding using non-parametric methods, we performed a Kruskal-Wallis test, and found that for five of the twelve solutions, at least two groups had a distinct distribution of ratings: 'Sustainability grants' (p = 0.00234), 'Help finding funding' (0.0356), 'A learning community' (0.0248), 'Mentoring programs' (0.0159), and 'Publicity' (0.0126).

Aspiring contributors were asked a similar question. They were asked which solutions would help them get started contributing, and could select all applicable options. For aspiring contributors, 'Computing environments' was the most popular solution (40 respondents (65.6%)), with 'A learning community' (37 respondents (60.7%)) and 'Educational materials' (35 respondents, (57.4%)) close behind (Fig 9C). Funding-related solutions were less popular among aspiring contributors: 'Dedicated grants' was selected by only 22 respondents (36.1%), and 'Help finding funding' by only 21 respondents (34.4%).

## Participants' comments on challenges and solutions

To capture challenges beyond those we anticipated, we asked contributors to write in any other challenges they've faced and/or types of support that they might find helpful. We received 41 informative responses, which revealed twelve themes including such topics as funding, open source education, lack of time, and the burden of open-source software maintenance (Fig 10). We grouped these twelve themes into three overarching categories: Culture, Resources, and Infrastructure. The emergence of the culture theme is particularly interesting because the survey instrument did not ask any questions about culture or policy. The most common theme under the Culture category was "University leadership, norms, and priorities," and some representative comments from this sub-theme are shown in Table 2. Other comments in the Culture category included calls for clearer guidance regarding open source policy, more education about open source and open source careers, and more appreciation for the value of open source. All comments and codes are available in Table 18 in S1 Data.

This question was shown to experienced open source contributors only. The prompt was, "Are there any other challenges you've encountered in open source, or types of support that you would find helpful?" A full list of comments and codes is available in Table 18 in S1 Data. Comments have been de-identified, including indirect identifiers such as projects or departments. Many comments contain typos or grammatical errors. These are original to the respondent and we have deliberately left them as-is.

**Comments on challenges and solutions**

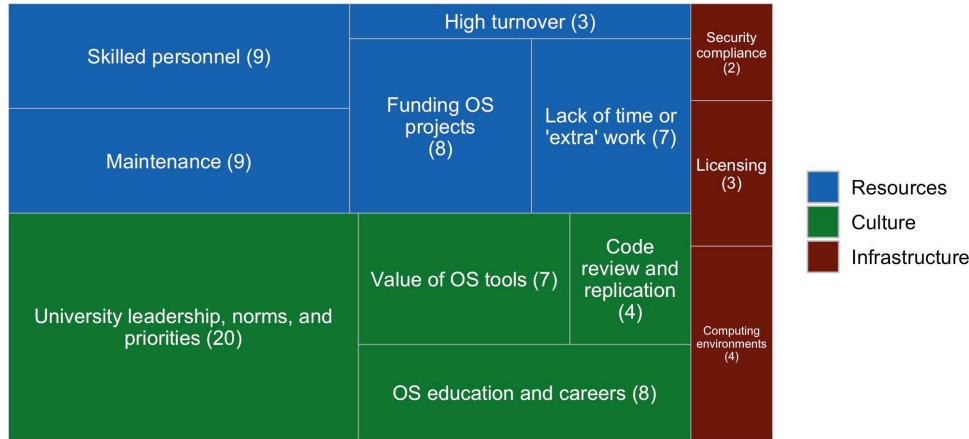

**Fig 10. Comments on challenges and solutions in open source.** 12 themes were found in comments from experienced contributors, and these 12 themes were grouped into three categories: Resources, Culture, and Infrastructure. The prompt was, "Are there any other challenges you've encountered in open source, or types of support that you would find helpful?" Numbers in parentheses indicate the number of comments that referenced that theme.

Seventeen solutions were proposed in the comments, including several that reinforced ideas from the survey as well as new and original suggestions. The additional solutions not included in the survey instrument are presented in Table 3 and include code review, education about open source careers, and more.

## Discussion

### Summary of findings

In this study, we present findings from self-reported contribution patterns of UC open source contributors, as well as opportunities for support. While a number of studies have focused on the needs and/or habits of academic developers of open-source research software [26,32,75–77], few have included perspectives from non-research staff. Rooted in the academic OSPO perspective, this study examines the university open source community as a whole, including both academics (students, teachers, and researchers) and non-research staff (IT staff, library staff, and more).

While the sample size was fairly small relative to the entire UC population (see Limitations), we believe several general conclusions may be drawn from our data set. Open source contributors, on average, have held four roles (e.g., Bug/Issue Reporter, Contributor, Maintainer), and have contributed to two to three different types of projects (e.g., applications, websites, or packages). More than half (58%) identified as open-source maintainers, outnumbering community managers, UI/UX designers, and IT/systems administrators by about three to one. This is higher than seen in previous surveys: Pinto et al. 2016 found that 51% of contributors to popular open source projects are either maintainers or core contributors [78], Nagle et al. 2021 found that 54% of open source contributors identified as maintainers on at least one project [79], and Hendrick and Ramaswami 2024 found that only 36% of contributors identified as either maintainers or core contributors [80]. Notably, these studies have all been conducted outside of academic contexts, therefore one possible explanation for our findings is that academic projects may be smaller on average, which could result in a higher proportion of maintainers. Another possibility is that due to our positionality in the UC community and our snowball sampling approach, our survey likely reached a large number of open source enthusiasts, which would include many maintainers. Nevertheless, we present evidence that a majority of university open source contributors have also served as project maintainers, and we believe that this finding should spur university decision makers to consider how they might support a high number of maintainers on their campuses.

We also found that while all groups contribute to small projects more than they contribute to large projects, non-research staff report contributing to large projects at a higher frequency than academics do. This is particularly striking because IT staff (a subset of non-research staff) were less likely than academics to say that open source work is part of their job. While we did not directly ask staff contributors whether those large-project contributions are part of their job, these findings hint at the possibility that many IT staff may be contributing to high-impact open-source projects, but doing so in addition to their regular job duties. Meanwhile, 73% of academic open source contributors report that they never or rarely contribute to large projects. This may be because academics primarily work with, and contribute to, specialized tools in their field rather than general-purpose infrastructural tools. These findings raise questions for the research university's mission and priorities, such as: Should non-research staff and/or researchers be rewarded for contributions to large, foundational projects? Should such contributions be considered part of their job duties? Do universities have a responsibility to, or would they benefit from, being strategic about their open source contributions?

GitHub was by far the most popular platform for sharing code and/or hardware designs, with 95% of contributors reporting having used it, followed by "a custom website" (31%). While sharing code and designs on personal GitHub repositories or websites can showcase skills and support open-source work, it also carries risks such as leaking sensitive data, violating employer or intellectual property policies, or failing to properly archive work in the scholarly record. Additionally, sharing the work through personal accounts or websites might make the work harder to find. This limits opportunities to engage contributors and effectively steward that work. Academic software users already face significant barriers to discovery of research software in repositories [81,82], and sharing code through personal websites likely exacerbates

this problem. Based on the data we analyzed for this project, it is still unclear what share of respondents use personal vs. institutional GitHub accounts, and to what extent they follow open source best practices. Although identifying trends from GitHub repositories is not trivial [36,37] and falls beyond the scope of this study, we anticipate the GitHub usernames obtained through our survey for outreach purposes could be used for such exploration.

The challenge open source contributors encounter most frequently is lack of time for documentation, coding, and self-education. This aligns with the finding from [32] that about three quarters of research software developers lack sufficient time to engage in available training, the finding from [31] that around two thirds of research staff who develop software feel inadequately trained, and the finding from [78] that the most common reason casual contributors do not become more engaged is lack of time. Challenges about funding and hiring were common in a different sense—relatively fewer participants identified themselves as responsible for addressing these challenges, yet those who did reported engaging with them frequently. The high rate of "non-applicable" responses to these challenges might be due a perception that open source work is inherently unpaid, or a perception that resourcing is the responsibility of management only, not more junior staff. The next-most common challenges were about managing issues and attracting users. These challenges relate to maintenance of an open-source codebase, and thus their prevalence is consistent with the high number of maintainers in our survey sample. Security and licensing, issues that are often among the corporate OSPO's core responsibilities [1], were ranked as less frequent challenges than time, funding, and education. This suggests that there is a balance to be struck between institutional risk management and providing services that alleviate contributors' primary pain points.

Preferred solutions varied by group, but some nearly-universal trends did emerge. The most popular solution among experienced open source contributors was "Dedicated grants for open-source project sustainability". These findings are echoed by Mourão et al. [30] who found that academic researchers highlight lack of funding and long term incentives as a challenge for research software success, and the URSSI survey results [32], in which the vast majority of developers surveyed reported that their institution did not provide adequate financial support for their research software development. Interestingly, that study also found that researchers may underrepresent software-related work in their funding proposals. As noted in [32], fewer than half of respondents reported including software development costs in grants, and even fewer accounted for maintenance or reuse. The survey did not explore why researchers may not account for these costs, but the results point to a gap between the recognized need for sustainable software funding and the institutional support available to software projects.

While the most popular solutions among most experienced contributor groups were related to funding, 'A learning community', 'Mentoring programs', and 'Educational materials and workshops' were more popular than funding-related solutions among aspiring contributors. Similarly, the UW-Madison survey found that 87% of respondents "would like to see more training, education, or support for learning how to contribute to open-source projects" [41]. These results suggest that while funding-related initiatives are likely to be widely appreciated by experienced contributors, educational solutions are more likely to be popular with those just getting started.

Interestingly, "Access to free, feature-rich computing environments" was the only solution that was in the top three for both experienced and aspiring contributors. Depending on how we calculated the ranking, this solution scored #2 or #3 among experienced contributors, second only to funding-related solutions. We believe this is an important finding that warrants further research. Possible reasons for the popularity of this solution include: the promise of "free" resources; users may simply be unaware of existing campus offerings; demand for GPUs from a surge in AI workflows may be outpacing campus cluster resources [83]; or demand may be shifting away from on-premises clusters and toward commercial cloud offerings [84,85]. Whatever the driving factor, it is evident that a large number of UC open source contributors feel constrained by the limitations of their existing compute resources.

Among experienced contributors, 'A learning community' scored 4th based on ratings, but 8th using means from the regression model. We suspect at least two reasons for this. First, weighting all the job categories equally in our model reduced the influence of non-research staff, as discussed in Results. Second, when tallying ratings, 'A learning community' was only

slightly ahead of 'Documentation help' and 'Legal support'. Given that our model incorporated several sources of variation that descriptive statistics do not capture, it is reasonable that there might be some reshuffling of solutions that were separated by only small differences. In summary, while 'A learning community' appears to be reasonably popular, particularly among non-research staff, its popularity was not as stable as that of 'Sustainability grants' or 'Computing environments'.

The qualitative comments also revealed crucial new insights. We received a large number of comments addressing university leadership, priorities, norms, and culture. We also found two unanticipated motivations for contributing to open source: expressing academic values and countering perceived exploitation from technology vendors. Open source values overlap with academic ideals such as collaboration, transparency, and autonomy [7–9,86]. Consistent with this, the UW-Madison survey [41] found that more than half of respondents viewed having a vibrant open source culture as "very valuable". This suggests that research universities stand to gain from recognizing open source as an expression of their mission and a contribution to their social impact. Meanwhile, some participants' frustration with technology vendors, paired with comments about a lack of appreciation for the value of open source, hint at a possible cultural schism wherein only those who are informed about open source appreciate its significance and impact. Studies have shown that many maintainers struggle with feeling underappreciated [87,88], a sentiment echoed by several comments in this survey. These comments, which emerged organically from open-ended prompts, reveal that the problem of supporting academic open source is not merely a technical issue, nor is it purely a matter of resources. To promote greater engagement with open source, university staff and leadership must foster a culture of open source contribution, acknowledge the value of open source work, and demonstrate their commitment to open source principles through policy and action.

## Limitations

This was the first effort to survey open source contributors across the UC system, and we learned many lessons along the way. One of the major threats to the validity of this study is our ad-hoc survey distribution approach. Our ability to distribute the survey was limited by our social networks, available campus communication channels, and our ability to secure access to those channels. As a result, nearly half of our respondents came from just two campuses, and only seven undergraduate experienced contributors completed the survey. It is possible that many, or even most participants found the survey through UC OSPO Network mailing lists and Slack channels, meaning that they are sufficiently invested in open source to be checking our announcements. This would lead to an overrepresentation of highly experienced contributors in our sample. Future deployments of this survey by our team will strive for greater homogeneity in our per-campus samples, ideally securing a UC-wide communication channel, so that the same message could reach all campuses by the same means.

Analysis of the survey results also revealed possible answers the survey team had not considered. For example, comments on the question about code-sharing platforms mentioned Software Heritage, package distribution services like CRAN and PyPi, and self-managed Git servers, options we will probably include in the future. Additionally, the discovery that time was the top challenge raises the question: how many people would have selected 'time' as a solution, had it been offered as an option (for example, as a reframing of job duties)? Since this option was not included, it is unclear whether the strong preference for sustainability grants comes from high operating costs per se, or whether respondents simply wish to 'buy time'. While no survey will ever be fully comprehensive, we expect that future iterations of this survey instrument will be informed by these and other results of this survey.

Finally, the conclusions reached in this study are from a large, public, research-intensive university system. More research is needed to understand whether our conclusions could apply to other types of university, including private research universities and primarily undergraduate institutions.

## Conclusions and future directions

This work is the first in-depth research initiative by the UC OSPO Network, and is a foundational study in the burgeoning body of research by academic OSPOs. Fruitful directions for future research include:

1. Deeper investigation into contributor needs with respect to computing infrastructure,

2. More research into which specific open-source projects universities are relying on and/or contributing to,

3. 4Replication of this survey at other universities, and

4. Lessons learned from implementations of pilot programs based on the findings presented here.

Our observations are not intended to be prescriptive, but rather to equip university decision-makers with data that can drive their strategic planning. Contributor needs must be balanced by consideration of institutional priorities and resources. Nevertheless, we recommend academic OSPOs and other support staff consider the following:

1. Time and money were identified as the most pressing concerns for university open source contributors. These are common limitations in academia, so support staff might need to think creatively about how to provide time and money in direct or indirect ways. For instance, communities of practice and co-working groups can motivate people to spend time on backburner tasks, and new AI tools may help speed up those tasks. Distributing grants and awards would likely be useful, but if resources are limited, there may be other approaches OSPOs could take, including highlighting external funding opportunities, helping projects source volunteers, assisting with grant writing, or helping maintainers design a more lightweight product. OSPOs could also advocate for systems that allow university personnel to devote a portion of their time to open source activities.

2. "Access to free, feature-rich computing environments" was the only solution that was rated highly with both experienced and aspiring contributors. While computational infrastructure is not typically a core function of the industry OSPO [1], perhaps there is an opportunity for academic OSPOs to collaborate on improving the state of campus computing resources.

3. While experienced contributors favored funding-related solutions, aspiring contributors favored educational solutions. This indicates that academic OSPOs should always be clear on their target audience, because different groups have different needs. If resources are limited, directing OSPO services toward one target group, at least initially, may be appropriate. The OSPO might then collaborate with other support teams to serve different groups.

4. When designing our survey, we undersampled the influences of ideology and culture in open source activity. Academic OSPOs or other support staff may consider setting goals with respect to "culture work": work that demonstrates the university's appreciation of both open source principles and the people who strive to enact them.

This study provides one of the first comprehensive views of the university open source contributor community, highlighting the distinct needs of students, researchers, and non-research staff. Despite nuances among the needs of different groups, our findings highlight a prominent struggle for sustainability. With nearly one in three experienced contributors identifying grants for software sustainability as the most useful thing the university could do to support their work, this study adds to the growing body of literature that indicates that current institutional investment is insufficient to meet the demands of open source work. Given the widespread usage and creation of open source software at research universities, universities should take an active role in stewarding the software they produce and depend on. OSPOs could help fill this gap, moving beyond resource discovery to establishing the support structures necessary to ensure the long-term viability of the open source tools that power modern research.

## Supporting information

**S1 File. Survey instrument.** Survey instrument in PDF format. See Materials and methods for a description of survey flow.
(PDF)

**S1 Fig. Composition of job categories by open source contribution experience.** (A) Total number of aspiring and experienced contributors, for each job category. (B) Percent of aspiring and experienced contributors, for each job category.
(PDF)

**S2 Fig. No correlation between campus size and response rate.** Scatterplot showing, for each UC campus, the approximate size of the campus community (staff and students, based on data from the UC Information Center), versus the number of responses received from that campus in this survey. The coefficient of determination for this relationship is shown in the top-right.
(PDF)

**S3 Fig. Maintainer/contributor/bug reporter overlap.** UpSet plot showing the number of survey respondents who identified as a Maintainer and/or Contributor and/or Bug reporter (n = 233 total respondents). The third column from the right shows respondents who identified with none of these three roles. Right-side bars show the total number of respondents who selected each role. Note that throughout this report, "contributor" is used as a broad umbrella term, but the term "contributor" for the purposes of this question is more specific, and is defined in the survey instrument Q4 as "Contributing relatively small amounts of code or hardware design, for example by fixing bugs or adding new features.".
(PDF)

**S4 Fig. Ordinal regression of participants' contribution frequency to projects of a certain size.** (A) Minimal model accounting only for variation between individuals. Reference level was arbitrarily selected to be 'Small'. (B) Model that also includes job category as a fixed effect and an interaction between job category and project size. Reference levels were arbitrarily selected to be 'Students' and 'Small'.
(PDF)

**S5 Fig. Frequency of contributions to large and small projects.** Y-axis shows the percent of eligible respondents in each job category who selected the x-axis value shown. The question was, "How frequently have you contributed to projects of the following size? For each project size, please answer relative to the other sizes.". (A) Large projects. (B) Small projects.
(PDF)

**S6 Fig. Contributions to different project types by job category.** Horizontal axis indicates the percent of respondents in each job category who reported having contributed to the project type on the vertical axis.
(PDF)

**S7 Fig. Usage of code/data sharing platforms.** (A) Percent of respondents in each field of study who reported having ever used the platform on the x-axis. (Limited to academics.) (B) Percent of respondents from each UC campus who reported having ever used the platform on the x-axis. (C) Percent of respondents from each UC campus who reported ever having used the data repository on the x-axis. Note that numbers do not necessarily add up to 100% because participants could have selected all or none of these options.
(PDF)

**S8 Fig. Top five challenges for each job category, according to two different methods, from data in our survey population.** (A) Mean rating after coding rating scale responses to numeric values (e.g., "Never" = 0, "Rarely" = 1, etc.). (B) Percent of respondents in each job category who selected "Frequently" or "Always".
(PDF)

**S9 Fig. Statistics demonstrating faculty's preferential rating of cluster 2 challenges.** (The cluster 2 challenges are 'Finding funding', 'Securing funding', and 'Hiring'.) (A) Output from ordinal regression of challenge frequency ratings on

job categories and cluster 2 challenges. Reference levels were arbitrarily selected to be "Faculty" and 'Finding funding'. (B) Estimated marginal means from regression in a). (C) Significantly different job category comparisons according to a Dunn's test, for each cluster 2 challenge.
(PDF)

**S10 Fig. Top five solutions for each job category, as observed in our survey population, according to several different methods.** (A) Mean rating after coding rating scale responses to numeric values ("Not very useful" = 0, "Useful" = 1, "Very useful" = 2). (B) Percent of respondents in each job category who selected "Useful" or "Very useful". (C) Top five solutions from Q11, where participants had to choose their favorite solution, in terms of the number of participants who chose that solution.
(PDF)

**S11 Fig. Regression of participants' ratings of solutions.** Participants rated the utility of each solution on a scale of "Not very useful", "Useful", "Very useful". (A) Output from minimal ordinal regression model of utility ratings on solutions, with participant as a random effect. Reference level was arbitrarily selected to be 'Computing environments'. (B) Output from ordinal regression of utility ratings on job categories and solutions, with an interaction between the two. Reference levels were arbitrarily selected to be "Faculty" and 'Computing environments'.
(PDF)

**S1 Data. Table 1. Academic participants' disciplines, classified at level one of the Digital Commons Taxonomy.** See Materials and methods for details. **Table 2. Academic participants' disciplines, classified at level two of the Digital Commons Taxonomy.** Disciplines with the same count are grouped together, e.g., two participants were from Biostatistics, two participants were from Computational Biology, etc. See Materials and methods for details. **Table 3. Academic participants' disciplines, classified at level three of the Digital Commons Taxonomy.** Disciplines with the same count are grouped together, e.g., one participant was from Aerospace engineering, participant was from Agriculture, etc. See Materials and methods for details. **Table 4. Importance of open source.** Results from Q2 about the importance of open source for various kinds of work. The first numerical column indicates the number of eligible participants who said that open-source software projects are either "Important" or "Very important" for a particular type of work. The "Total Participants" column indicates the total number of eligible participants. For the teachers and researchers groups, eligible participants are those who did not select N/A. For the students group, eligible participants did not select N/A and are graduate or undergraduate students (as indicated in Q16). For the non-research staff group, eligible participants are those who did not select N/A and are non-research staff (Q16). **Table 5. Contributor role counts.** "Role" indicates the role described on the survey (S1 File Q4). "Count" indicates the number of participants who selected that role. Participants could select all that apply. **Table 6. Project sizes by job category.** "N small" and "Percent small" indicate the number/ percent of people who contribute to small projects "Occasionally" or "Relatively Frequently". "N large" and "Percent large" indicate the number/percent of people who contribute to large projects "Occasionally" or "Relatively Frequently". **Table 7. Contribution frequencies by project size.** Estimated means from ordinal regression. Note that these estimates are on the "mean.class" scale, which in this case is 1–4 (see Materials and methods). "asymp.LCL" and "asymp.UCL" are asymptotic lower and upper confidence limits. See emmeans package documentation for more details. **Table 8. Pairwise differences between estimated mean contribution frequencies, for each project size within each job category.** Here, "Estimate" refers to the difference between the means. P-values were adjusted for multiple comparisons using the default Tukey method. "asymp.LCL" and "asymp.UCL" are asymptotic lower and upper confidence limits. See emmeans package documentation for more details. **Table 9. Pairwise differences between job categories' estimated mean contribution frequencies, for each project size.** Here, "Estimate" refers to the difference between the means. P-values were adjusted for multiple comparisons using the default Tukey method. "asymp.LCL" and "asymp.UCL" are asymptotic lower and upper confidence limits. See emmeans package documentation for more details. **Table 10. Frequency of**

contributions to small vs. large projects, by job category.** Results of paired, one-sided (frequency of small > frequency of large) Wilcoxon signed-rank tests within each job category. "(Pseudo)median" indicates estimated median increase (Hodges-Lehmann) from frequency of contributions to large projects to frequency of contributions to small projects. "CI Lower" indicates the lower bound on the 95% confidence interval, and p-value indicates the p-value, adjusted for multiple tests using the "holm" method implemented in the stats package. **Table 11. Project type counts.** "Project type" indicates the project type described on the survey. "Count" indicates the number of participants who indicated having ever contributed to a project of that type. Participants could select all that apply. **Table 12. Comments from Q6 about motivations for contributing to open source.** "Content" column contains the comments from the survey. Other columns are codes from coding analysis. An "X" indicates that that code was applied to that comment. For additional privacy, the order of the rows has been randomized. **Table 13. Descriptive statistics from Q9 about challenges.** "Total" indicates the total number of "points" garnered by that challenge. Each participant contributed 0–4 points per challenge based on their response, as follows: "Never" = 0, "Non-applicable" = 0, "Rarely" = 1, "Occasionally" = 2, "Frequently" = 3, "Always" = 4. Other statistics are from the distribution of points scores for each challenge. **Table 14. Descriptive statistics of solution ratings.** Participants were asked to rank solutions by their utility, and responses were coded to numbers like so: Non-applicable = 0, Not very useful = 0, Useful = 1, Very useful = 2. The rating values were then summed for each solution to produce the "Total" column. All other columns are descriptive statistics on the distribution of rating values. **Table 15. Estimated marginal mean solution ratings, holding job constant, weighting job categories equally.** Estimates are from an ordinal regression model that models outcomes (rating) as probability distributions on a scale of 1–3. Job category groups are weighted equally (the default in the emmeans package). "asymp.LCL" and "asymp.UCL" are asymptotic lower and upper confidence limits. See emmeans package documentation for more details. **Table 16. Estimated marginal mean solution ratings, holding job constant, weighting job categories proportional to sample size.** Estimates are from an ordinal regression model that models outcomes (rating) as probability distributions on a scale of 1–3. Job category groups are weighted proportional to their sample size, so larger groups have more influence on the mean rating. "asymp.LCL" and "asymp. UCL" are asymptotic lower and upper confidence limits. See emmeans package documentation for more details. **Table 17. Pairwise differences between estimated mean solution ratings, for each job category.** Here, "Estimate" refers to the difference between the means (means not shown). For example, Faculty rated 'Computing Environments' 2.353, and they rated 'Publicity' 2.1117, so the difference is 0.236. P-values were adjusted for multiple comparisons using the default Tukey method. "asymp.LCL" and "asymp.UCL" are asymptotic lower and upper confidence limits. See emmeans package documentation for more details. **Table 18. Comments from Q12 about additional challenges and/or solutions.** "Content" column contains the comments from the survey. Other columns are codes from coding analysis. An "X" indicates that that code was applied to that comment. For additional privacy, the order of the rows has been randomized. Uninformative responses such as "?" or "N/A" have been removed. Comments have been de-identified, including indirect identifiers such as projects or departments. Many comments contain typos or grammatical errors. These are original to the respondent and we have deliberately left them as-is.
(XLSX)

## Acknowledgments

The authors thank all the pre-test respondents who provided feedback on the survey instrument, and all survey respondents for their time. The authors also thank Stephanie Lieggi, Vessela Ensberg, Karla Padilla, David Minor, Kirstie Whittaker, and Todd Grappone for their assistance distributing and advertising the survey.

## Author contributions

**Conceptualization:** Renata Gonçalves Curty, Greg Janée, Amber E. Budden.

**Data curation:** Virginia T. Scarlett.

**Formal analysis:** Virginia T. Scarlett.

**Funding acquisition:** Amber E. Budden.

**Investigation:** Virginia T. Scarlett, Renata Gonçalves Curty, Juanita Gomez, Laura Langdon.

**Methodology:** Virginia T. Scarlett, Renata Gonçalves Curty, Juanita Gomez, Laura Langdon, Greg Janée, Amber E. Budden.

**Project administration:** Virginia T. Scarlett, Amber E. Budden.

**Software:** Virginia T. Scarlett.

**Supervision:** Amber E. Budden.

**Validation:** Virginia T. Scarlett.

**Visualization:** Virginia T. Scarlett.

**Writing – original draft:** Virginia T. Scarlett, Juanita Gomez.

**Writing – review & editing:** Renata Gonçalves Curty, Juanita Gomez, Laura Langdon, Greg Janée, Amber E. Budden.

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
