## [Decision Letter · Decision Letter 0]

18 Feb 2026

PONE-D-25-68889A system-wide snapshot: A multi-campus survey of open source contributors at the University of CaliforniaPLOS One

Dear Dr. Scarlett,

Thank you for submitting your manuscript to PLOS ONE. After careful consideration, we feel that it has merit but does not fully meet PLOS ONE’s publication criteria as it currently stands. Therefore, we invite you to submit a revised version of the manuscript that addresses the points raised during the review process.

The three reviewers provide excellent feedback for your analysis. In particular, it would be good to position your study in the broader literature on the subject, to clarify or improve some of your statistical analysis and finally consider limitations of your study that result from the sampling procedure.

We look forward to receiving your revised manuscript.

Kind regards,

Vincent Antonio Traag, Ph.D.

Academic Editor

PLOS One

Journal Requirements:

2.  We notice that your supplementary figures are uploaded with the file type 'Figure'. Please amend the file type to 'Supporting Information'. Please ensure that each Supporting Information file has a legend listed in the manuscript after the references list.

Reviewers' comments:

Reviewer's Responses to Questions

**Comments to the Author**

1. Is the manuscript technically sound, and do the data support the conclusions?

Reviewer #1: Partly

Reviewer #2: Yes

Reviewer #3: Yes

2. Has the statistical analysis been performed appropriately and rigorously? 

Reviewer #1: I Don't Know

Reviewer #2: Yes

Reviewer #3: Yes

3. Have the authors made all data underlying the findings in their manuscript fully available?

Reviewer #1: No

Reviewer #2: Yes

Reviewer #3: Yes

4. Is the manuscript presented in an intelligible fashion and written in standard English?

Reviewer #1: Yes

Reviewer #2: Yes

Reviewer #3: Yes

5. Review Comments to the Author

Reviewer #1: Overall, I am interested in this work, as I work in this field and have also performed some similar work (which is in the process of being written up.) I would like to see a revised version of this paper published. My comments on it are:

1. While the survey was performed for the reasons discussed by the authors in the abstract, the results are also useful to those who study the open source and research software more widely, and can be used to compare with the results of other surveys.

Some examples:

[1] Brown A., Crouch S., Graham J., Grylls P.J., Hettrick S.J., Mangham S.W., Robinson J. & Wyatt C.”Research software at the University of Southampton”. 10 December 2019. http://eprints.soton.ac.uk/id/eprint/504695

[2] Mourão, E., Trevisan, D., & Viterbo, J. ”Understanding the success factors of

research software: interviews with Brazilian computer science academic

researchers.” In International Conference on Information Technology & Systems

(pp. 275-286). Cham: Springer International Publishing. 2023, February

[3] Besser, S. A., Jensen, E. A., & Katz, D. S. (2025). Research Software at the University of Illinois Urbana-Champaign: A Mixed Methods Survey Dataset (1.0) [Data set]. Zenodo. https://doi.org/10.5281/zenodo.15161372 (though no paper about this has yet been published)

I think it would be useful to frame the work, at least partially, in this larger scholarly context.

2. I think some discussion of the distinction between open source software and research software would be useful to add to the introduction. While it's clear that open source is the focus, it's unclear how much of this is research software vs more general open source, whether used for research or education or administration. Some of this is discussed in the figures and the discussion, but it would be useful to introduce this earlier in the paper.

3. Other sources of data exist that can be used to find open source information for a campus or system, such as GitHub and Software Heritage. It might be useful to mention these as well, even if collecting data from them and using this to enhance the understanding of the OS work at UC is beyond the scope of a revision.

4. The number of responses is extremely low, given the size of the community being surveyed. As a comparison, there were about 250 responses in [1] from a single university. This leads to a large question about selection bias. This is discussed in Limitations, but probably should be highlighted as a possible issue much earlier in the paper.

5. As a comment for PLOS more than for the authors, reading a draft that discusses figures and includes their captions but puts the figures themselves at the end (rather than in-line) is extremely frustrating and make the review harder and worse.

6. The DOIs listed for data and code availability don't resolve, so that the data and code are not currently available, and I was unable to check their completeness and quality.

7. I do not have appropriate expertise to judge the statistical analysis aspect of the work.

Reviewer #2: In this manuscript, Scarlett et al. report and discuss the results of a survey on UC campuses on Open Source contributions. The paper is timely given how much of modern society's infrastructure relies on software and how little is known about Open Source contributors dynamics in leading thought institutions globally.

Overall, the study is interesting, the methodology is mostly sound, and the manuscript is well written. I found minor statistical details that might need some clarification. In terms of data interpretation and discussion, I think the authors are a little too cautious. I do appreciate that they want their conclusions supported by data as much as possible, but I think that in a paper like this, part of the "Discussion" can include slightly more speculative angles that might give perspective to the individual findings. For instance, the fact that many contributors report lacking time rather then resources but then request resources (grants) to close that gap signals a clear inconsistency in my mind, however this is not directly discussed in the current manuscript.

Another point I'd like to raise is the current lack of focus on gender issues, a known strong bias in many open source communities. I wonder whether the authors might want to comment about this specific issue.

In conclusion, I think that this work is useful and well thought through. I add individual comments below that hopefully will help clarify specific items throughout the draft. They should be taken as nonbinding suggestions rather than hard requirements.

- Line 100: Unclear whether that first survey is this paper?

- Line 195: Complete list of packages not in references: this is part of the issue isn't it. This paper is not expected to fix this but at least acknowledge it.

- Line 199: Why is the job a fixed effect for all analyses? I get it that it's most likely the right thing to do, but it would be useful to have a couple passes without just to get a sense of the data with no filters.

- Line 235: I'm not sure I follow. The job categories are made somewhat arbitrarily (and further coarse grained occasionally), therefore the most obvious thing to do would be to not touch the data instead of rebalancing according to said arbitrary categories. The impression that this is "equal weighting" seems misguided to me. The real issue, of course, is that noise from rarely populated categories is going to pollute everything else.

- 251: Not sure what the power analysis is supposed to tell me since the 80% and 0.05 are arbitrary thresholds, but I guess it does not hurt to have this?

Results:

- 314: How does that compare to roughly estimated demographics in those campuses? To other surveys?

- 317: Part of the appeal/success of OSS technology is that it cuts through traditional disciplinary boundaries. I wonder how people with activities across multiple "fields" would feed about this categorisation.

- 322 and surrounding: Following up on the previous points, I wonder how a bioinformatician or chemical physicist / physical chemist would feel about this breakdowns.

- I also wonder how these percentages relate to total population numbers in those fields on campus. Medicine employs a lot of people in some campuses and therefore might have an outsized effect on the percentages. Without these baseline references, the percentages alone are a little confusing to me.

- 327 etc: Same as above, how do these percentages relate to total campus / staff numbers? I wonder whether UCs employ a lot of librarians (including part-time?) or whether librarians are more invested in Open Source.

- 343: How many actually said "not applicable"? It sounds like a small percentage, but it might be good to just write it out.

- 365: Phrasing is a little odd here.

- 390: Some comment on whether any of this was expected a priori would be useful I think.

- 452: I guess no multiple hypothesis correction here?

- 468: Same.

- 518: I think many OS contributors would assume - given the current funding landscape for OS projects - that "Resourcing" is out of the picture to start with, i.e. that OS development is intrinsically an unpaid, volunteer activity. So I don't know that I agree with the authors' interpretation that "Resourcing is not perceived as widespread", or at least I find it a little simplistic and misleading as currently phrased. I would add a comment of sorts somewhere in the paper (Discussion?).

- 527 and following: Indeed, this confirms my last comment. Faculty at least seems to know there is the *possibility* of funding - I imagine because they are used to applying for grants all the time anyway - and therefore has an easier time identifying Resourcing as even an issue, whereas many other categories might find lack of resources an "immutable" state of affairs and therefore, as a natural phenomenon, not a "challenge". I think adding a comment in this direction would help the readers reason on what might be going on, even though it is a little speculative.

- 541: This is a little awkward I think. Most people seem to say they lack primarily time, but when asked for solution they do not request time but resources instead. Of course, money can buy a developer's time to some degree, but I think this might speak to the fact that if OS contributions were considered a prestigious / legitimate activity in academia, then people might actually request for time instead of funding. In other words, I suspect that if UC campuses adopted the Google policy of yore "one day a week do whatever you are passionate about", then the answer to this question might change drastically. I am digressing, but I think a mention of this inconsistency somewhere in the paper would be useful.

- 587: I wonder how much of this is due to the AI bubble, which requires very special computing environments. New contributors might be most excited about this fashionable field and therefore be feeling this need more acutely. Perhaps adding a comment somewhere?

Other questions:

- Was there no question about age or age brackets? It would be interesting to see when in life people tend to get involved for what roles (mainteiner, bug reporter, etc.).

- Also, it would be interesting for future work to find a way to ask how people deeply enmeshed in open source finally stop their contributions. OS burnout is a serious issue and the authors might have a good platform to study this question quantitatively.

Reviewer #3: This manuscript presents findings from a multi-campus survey (n = 294) conducted across the University of California system to characterize open source contributors and identify their needs from the perspective of an academic Open Source Program Office (OSPO). The study integrates quantitative survey analysis (including ordinal regression, clustering, and non-parametric tests) with qualitative thematic coding. The authors report that open source is critical to academic work, that a surprisingly high proportion of respondents have served as maintainers (58%), and that lack of time—particularly for documentation—along with funding constraints and institutional culture are the most frequently reported barriers. The authors conclude with policy-relevant recommendations for supporting sustainability within academic OSPO initiatives.

While the study is limited to a single institution, it can be used as a good template for similar studies in other universities. It aligns with PLOS ONE's scope, the manuscript is generally sound, though several clarifications and refinements would improve its claims and generalizability.

1. The use of non-probability snowball sampling is clearly described, but the implications for representativeness are not sufficiently emphasized in the Discussion. Given that recruitment channels likely reached individuals already engaged in open source communities, the sample may overrepresent highly active contributors, in particular maintainers.

The claim that 58% of contributors have served as maintainers is striking and framed as a novel finding. However, without a defined sampling frame or baseline institutional estimates, it is difficult to interpret whether this proportion reflects the broader UC open source population.

Also the text do not make it clear on how a "maintainer" is defined. Is it the user with highest amount of contributions, owner of the repository, self-defined?

2. The manuscript occasionally shifts from describing UC-specific findings to discussing broader "universities" or "the academy". While the policy implications are broadly relevant, the empirical base is confined to one university system. Given UC's size, structure, and research intensity, it may not be representative of smaller or differently structured institutions. I recommend the authors to avoid generalizing or extrapolating the findings to other institutions. However, the authors should provide and suggest that studies of different universities could be conducted with the proposed framework.

Minor points:

Clarify distinctions between "experienced contributors", "aspiring contributors", and "users". Ensure consistent use of "non-research staff" vs "IT staff" vs "academic".

The emergence of cultural themes is one of the more interesting findings. Consider expanding this section slightly, as it appears organically from qualitative data instead of being prompted.

Some figure-heavy sections could benefit from slightly more synthesis in the main text to prevent fragmentation. Also, ensure all key claims are explicitly tied to figure numbers.

The Results section is potentially too dense. The Discussion could be slightly tightened to avoid repetition of detailed numeric results already presented.

Given the comments above, I recommend that the manuscript be revised before being considered for publication.

6. PLOS authors have the option to publish the peer review history of their article (what does this mean?). If published, this will include your full peer review and any attached files.

Reviewer #1: **Yes:**Daniel S. Katz

Reviewer #2: **Yes:**Fabio Zanini

Reviewer #3: **Yes:**Filipi Nascimento Silva

---

## [Author Response · Author response to Decision Letter 1]

31 Mar 2026

Dear Editor and Reviewers,

Thank you for your thoughtful feedback on our manuscript. We deeply appreciate the time and expertise you have invested in reviewing our work. Your comments have been instrumental in improving the clarity and rigor of our paper. Here is a high-level summary of the revisions:

• We have added several new literature references and tightened our definitions of key terminology.

o (Note that our citation manager does not work on a document that contains tracked changes, so newly added references are not properly integrated in the tracked changes version. References are properly integrated only in the final version.)

• We have added more discussion around the limitations of our snowball sampling method, ensuring that our claims about contributor demographics (like the 58% maintainer statistic) are properly contextualized.

• We have added minimal regression models to our supplemental materials where appropriate, and also explained our rationale for using job categories as a fixed effect. We have also added a supplementary data table showing mean solution ratings with an alternative weighting scheme.

• We have clarified the purpose of our demographic analysis and, where possible, referenced publicly available demographic data from UC, including adding a new supplementary figure.

• In this letter, we explain our rationale for minimizing the collection of personally identifiable information (PII), including indirect identifiers, such as gender and age, focusing only on data directly relevant to the study and keeping the survey centered on actionable OSPO services.

• We have embraced the reviewers’ invitation to be slightly more speculative in our Discussion, particularly by exploring the inconsistency between contributors reporting a lack of time but requesting financial resources. We have also expanded our commentary on the cultural themes that emerged organically from the qualitative data.

• We have refined phrasing throughout the manuscript and implemented a few minor typographical corrections that we ourselves spotted during the revision process.

Below, we detail our responses to each of the points raised, grouped thematically for your convenience. We have labeled each comment using the following nomenclature: E.1 = Editor comment 1; O.1 = author-initiated comment 1, R2.3 = Reviewer 2, comment 3, etc. We have reproduced all comments with their labels here, along with our responses. Comments that refer to a particular passage in the text are also referenced in the tracked changes on the manuscript.

I. Journal Formatting and Data Availability

• [E.1] Style Requirements: “Please ensure that your manuscript meets PLOS ONE's style requirements, including those for file naming. The PLOS ONE style templates can be found at https://…”

o We confirm that the manuscript and file names are correctly formatted according to PLOS ONE’s style templates.

• [E.2] Supplementary File Types: “We notice that your supplementary figures are uploaded with the file type ‘Figure’. Please amend the file type to ‘Supporting Information’. Please ensure that each Supporting Information file has a legend listed in the manuscript after the references list.”

o We will amend the file type of the supplementary figures to ‘Supporting Information’ in the submission portal. As requested, legends are already located at the end of the manuscript.

• [E.3] Recommended Citations: “If the reviewer comments include a recommendation to cite specific previously published works, please review and evaluate these publications to determine whether they are relevant and should be cited.”

o We have evaluated and incorporated the reviewer-recommended citations.

• [E.4] Reference List: “Please review your reference list to ensure that it is complete and correct. If you have cited papers that have been retracted, please include the rationale for doing so...”

o We have reviewed our reference list. It is complete and correct, and to our knowledge, we have not cited any retracted papers.

• [R1.5] Figures at the End: “[A comment for PLOS, not for the authors] Reading a draft that discusses figures and includes their captions but puts the figures themselves at the end (rather than in-line) is extremely frustrating and make the review harder and worse.”

o We agree. Also, this format makes preprinting more onerous because the PLOS format is not suitable for a preprint. We formatted it this way to adhere to PLOS guidelines.

• [R1.6] Data/Code DOIs Not Resolving: “The DOIs listed for data and code availability don't resolve, so that the data and code are not currently available, and I was unable to check their completeness and quality.”

o As far as we are aware, we provided the private Dryad link for peer reviewers in the PLOS ONE submission form. We used the real DOI (which is still in draft mode and thus does not resolve yet) in the manuscript text because we wanted to ensure the correct DOI was in place for publication, avoiding the risk of accidentally publishing the private peer review link. Here is the peer reviewer link: http://h52wvrds.r.us-west-2.awstrack.me/L0/http:%2F%2Fdatadryad.org%2Fshare%2FLINK_NOT_FOR_PUBLICATION%2FgBoNHKBsC4-SmVbLyuMd4jlEAUH7ADOiEzBcyiMF8nA/1/0101019d2bcec164-7c24361c-ac6d-42ca-ae73-5f91be2719e3-000000/xf7Ln2wNARBfM9fhxcKAGYHbml8=472

II. Scholarly Context, Literature, and Terminology

• [R1.1] Larger Scholarly Context: “While the survey was performed for the reasons discussed by the authors in the abstract, the results are also useful to those who study the open source and research software more widely, and can be used to compare with the results of other surveys. Some examples: Brown et al. 2019… I think it would be useful to frame the work, at least partially, in this larger scholarly context.”

o We agree that framing the work within the broader context is valuable. We have added new mentions of the suggested papers and others at line 81. We have also incorporated findings from these references into the Discussion at lines 797 and 814. Finally, we added a sentence to the end of the abstract to underscore the paper’s broader significance.

• [R1.2] Distinction Between Open Source and Research Software: “I think some discussion of the distinction between open source software and research software would be useful to add to the introduction. While it’s clear that open source is the focus, it’s unclear how much of this is research software vs more general open source, whether used for research or education or administration. Some of this is discussed in the figures and the discussion, but it would be useful to introduce this earlier in the paper.”

o We have added clarification regarding this distinction to the Introduction at line 112.

• [R1.3] Alternate Data Sources (GitHub/Software Heritage): “Other sources of data exist that can be used to find open source information for a campus or system, such as GitHub and Software Heritage. It might be useful to mention these as well, even if collecting data from them and using this to enhance the understanding of the OS work at UC is beyond the scope of a revision.”

o We already mentioned these sources in the Discussion (around line 790), but we have added mentions of existing web-scraping literature (including a publication from the UC OSPO Network) at line 81.

• [R3.3] Terminology Distinctions: “Clarify distinctions between ‘experienced contributors’, ‘aspiring contributors’, and ‘users’. Ensure consistent use of ‘non-research staff’ vs ‘IT staff’ vs ‘academic’.”

o We have added clarification regarding “experienced contributors” around line 332. We believe the term “users” is already clear and did not want to dwell on it, as users are not the focus of the paper. We had already defined “aspiring contributors” around line 333. We were careful with the term “academic” and already defined it as students, teachers, and researchers in five distinct places. Finally, we have added clarifications on lines 545 and 766 to emphasize the meaning of IT staff, which is also explained in the demographics section.

• [R2.3] Survey Clarification: “Line 100: Unclear whether that first survey is this paper?”

o We have adjusted the grammar around line 125 to clarify that the survey mentioned is indeed the one discussed in this paper.

III. Sampling, Representativeness, and Demographics

• [R1.4, R3.1a] Low Response Rate & Snowball Sampling Bias: “The number of responses is extremely low, given the size of the community being surveyed. As a comparison, there were about 250 responses in [1] from a single university. This leads to a large question about selection bias. This is discussed in Limitations, but probably should be highlighted as a possible issue much earlier in the paper.”, “The use of non-probability snowball sampling is clearly described, but the implications for representativeness are not sufficiently emphasized in the Discussion.”

o We acknowledge the limitations of our sampling method. We have added new text at the end of the Introduction (line 135) and the beginning of the Discussion (line 737) pointing to the limitations section. Further comments about selection bias have been added around lines 750 and 892.

• [R3.1b] 58% Maintainer Claim Context: “The claim that 58% of contributors have served as maintainers is striking and framed as a novel finding. However, without a defined sampling frame or baseline institutional estimates, it is difficult to interpret whether this proportion reflects the broader UC open source population.”

o We have added comments in the second paragraph of the Discussion (line 743) to contextualize this finding in light of prior studies and our sample population.

• [R3.2] Extrapolating to Other Institutions: “Given UC’s size, structure, and research intensity, it may not be representative of smaller or differently structured institutions. I recommend the authors to avoid generalizing or extrapolating the findings to other institutions.”

o We have added comments at the end of the Limitations section (line 906) and replaced several references to “universities” with “research universities” or “UC” to avoid over-generalizing.

• [R2.8, R2.11, R2.12] Baseline Campus Demographics: “314: How does that compare to roughly estimated demographics in those campuses? To other surveys?”, “I also wonder how these percentages relate to total population numbers in those fields on campus. Medicine employs a lot of people in some campuses and therefore might have an outsized effect on the percentages. Without these baseline references, the percentages alone are a little confusing to me.”, “327 etc: Same as above, how do these percentages relate to total campus / staff numbers? I wonder whether UCs employ a lot of librarians (including part-time?) or whether librarians are more invested in Open Source.”

o Comparing our data to campus population data is challenging for a number of reasons, but we have incorporated public data where possible:

• We have noted that there is no correlation between campus size and the number of responses (line 358), and added a new supplementary figure (S2 Fig) demonstrating this.

• We included a “baseline” for the social sciences (line 369), but the remaining academic disciplines tracked by UC are not comparable to ours.

• Only a small fraction of UC’s workforce data are publicly available, making comparison impossible. A comment to this effect has been added at line 386.

o We have also added an explanation regarding the purpose of the demographic data at the top of the Demographics section (line 328).

• [R2.9, R2.10] Disciplinary Breakdown (Multidisciplinary Fields): “317: Part of the appeal/success of OSS technology is that it cuts through traditional disciplinary boundaries. I wonder how people with activities across multiple “fields” would feed about this categorisation.”, “322 and surrounding: Following up on the previous points, I wonder how a bioinformatician or chemical physicist / physical chemist would feel about this breakdowns.”

o We have added a comment regarding the inherent difficulties of disciplinary classification for roles like bioinformaticians or chemical physicists at line 374.

IV. Methodology and Statistical Analysis

• [R1.7] Statistical Expertise: “I do not have appropriate expertise to judge the statistical analysis aspect of the work.”

o Understood.

• [R2.5] Job as a Fixed Effect: “Why is the job a fixed effect for all analyses? I get it that it’s most likely the right thing to do, but it would be useful to have a couple passes without just to get a sense of the data with no filters.”

o We have updated two of the four regression sections to include a minimal model without this fixed effect variable. For the project size analysis, we added the minimal model to the main text and Supplemental Figure S4 (line 446). For the solutions analysis, we added the minimal model to S11 Fig (formerly S10) to aid in interpreting the more complex model. We left the motivations and challenges analyses as-is, first because we feel that in those cases a simple regression would complicate the existing exploratory analysis without adding value, and second because there is no corresponding minimal, nested model for those analyses (or none that makes sense, anyway).

• [R2.6] Arbitrary Job Categories: “The job categories are made somewhat arbitrarily (and further coarse grained occasionally), therefore the most obvious thing to do would be to not touch the data instead of rebalancing according to said arbitrary categories. The impression that this is ‘equal weighting’ seems misguided to me. The real issue, of course, is that noise from rarely populated categories is going to pollute everything else.”

o We have included a theoretical justification for these categories in the Methods around line 234. Furthermore, while somewhat arbitrary, these categories were significant sources of variation in most cases. We examined the job category variable in all regressions, but we only kept it where it improved model fit. (We have updated the text on line 226 from “all” to “most” to clarify this). We did not incorporate any weighting into the regression analysis, whose outputs are shown in S11 Fig (previously S10). It was only during the estimation of marginal means that we introduced a weighting scheme. We had already commented on a key outcome difference between the equally weighted and proportionally weighted EMMs, but we understand that choosing just one weighting scheme is opinionated, so we have now added the proportionally-weighted EMMs as a supplementary data file (S16 Data), so that readers may see the two complementary views of the data. Given that there were significant differences between job categories, we chose to model and then control for that variable rather than ignore it. If readers want a simple, minimally processed view of the data, they can refer to the descriptive statistics.

• [R2.7] Power Analysis Thresholds: “251: Not sure what the power analysis is supposed to tell me since the 80% and 0.05 are arbitrary thresholds, but I guess it does not hurt to have this?”

o While 80% and 0.05 are arbitrary thresholds, they are widely used standard thresholds.

• [R2.16, R2.17] Multiple Hypothesis Correction: “452: I guess no multiple hypothesis correction here?”, “468: Same. [No multiple hypothesis correction here?]”

o We did perform multiple hypothesis correction here but we forgot to mention it in the text. This has now been added to the Methods section (line 252).

V. Focus and Scope (Gender, Age, Burnout)

• [R2.2, R2.22, R2.23] Project Scope: “Another point I'd like to raise is the current lack of focus on gender issues, a known strong bias in many open source communities. I wonder whether the authors might want to comment about this specific issue.”, “Was there no question about age or age brackets? It would be interesting to see when in life people tend to get involved for what roles (maintainer , bug reporter, etc.).” “Also, it would be interesting for future work to find a way to ask how people deeply enmeshed in open source finally stop their contributions. OS burnout is a serious issue and the authors might have a good platform to study this question quantitatively.”

o We chose not to ask about gender, age, or other protected characteristics

---

## [Decision Letter · Decision Letter 1]

23 Apr 2026

A system-wide snapshot: A multi-campus survey of open source contributors at the University of California

PONE-D-25-68889R1

Dear Dr. Scarlett,

We’re pleased to inform you that your manuscript has been judged scientifically suitable for publication and will be formally accepted for publication once it meets all outstanding technical requirements.

Kind regards,

Vincent Antonio Traag, Ph.D.

Academic Editor

PLOS One

Additional Editor Comments (optional):

Reviewers' comments:

Reviewer's Responses to Questions

**Comments to the Author**

1. If the authors have adequately addressed your comments raised in a previous round of review and you feel that this manuscript is now acceptable for publication, you may indicate that here to bypass the “Comments to the Author” section, enter your conflict of interest statement in the “Confidential to Editor” section, and submit your "Accept" recommendation.

Reviewer #1: (No Response)

Reviewer #2: All comments have been addressed

2. Is the manuscript technically sound, and do the data support the conclusions?

Reviewer #1: Yes

Reviewer #2: Yes

3. Has the statistical analysis been performed appropriately and rigorously? 

Reviewer #1: I Don't Know

Reviewer #2: Yes

4. Have the authors made all data underlying the findings in their manuscript fully available?

Reviewer #1: Yes

Reviewer #2: Yes

5. Is the manuscript presented in an intelligible fashion and written in standard English?

Reviewer #1: Yes

Reviewer #2: Yes

6. Review Comments to the Author

Reviewer #1: In general, I am happy with the responses to my and other reviewer's comments.

Some small remaining issues:

I don't think "a preference for" in line 444 is an accurate description, as this implies motive.

In lines 545 and 764, I wonder if any IT staff are also research staff, as is the case in some academic institutions.

I've marked this as minor revision, but I do not feel a need to review this again just for these changes.

Reviewer #2: The authors have addressed all requested changes, mostly in a satisfactory fashion. Occasional disagreements, for instance on whether full package reference citation is or is not best practice, might be unresolved (it is best practice as far as I am concerned: recursive software dependencies are very real) but should not impact the overall high quality of the study nor its ability to proceed to acceptance.

Great job, a timely study.

7. PLOS authors have the option to publish the peer review history of their article (what does this mean?). If published, this will include your full peer review and any attached files.

Reviewer #1: **Yes:**Daniel S. Katz

Reviewer #2: **Yes:**Fabio Zanini

---

## [Editor Report · Acceptance letter]

PONE-D-25-68889R1

PLOS One

Dear Dr. Scarlett,

I'm pleased to inform you that your manuscript has been deemed suitable for publication in PLOS One. Congratulations! Your manuscript is now being handed over to our production team.

Kind regards,

on behalf of

Dr. Vincent Antonio Traag

Academic Editor

PLOS One